# Yeast Rad52 is a homodecamer and possesses BRCA2-like bipartite Rad51 binding modes

Jaigeeth Deveryshetty[1,6], Rahul Chadda[1,6], Jenna R. Mattice[2], Simrithaa Karunakaran[1], Michael J. Rau [3], Katherine Basore[3], Nilisha Pokhrel[4,5], Noah Englander[1], James A. J. Fitzpatrick [3], Brian Bothner [2] & Edwin Antony [1] ✉

Homologous recombination (HR) is an essential double-stranded DNA break repair pathway. In HR, Rad52 facilitates the formation of Rad51 nucleoprotein filaments on RPA-coated ssDNA. Here, we decipher how Rad52 functions using single-particle cryo-electron microscopy and biophysical approaches. We report that Rad52 is a homodecameric ring and each subunit possesses an ordered N-terminal and disordered C-terminal half. An intrinsic structural asymmetry is observed where a few of the C-terminal halves interact with the ordered ring. We describe two conserved charged patches in the C-terminal half that harbor Rad51 and RPA interacting motifs. Interactions between these patches regulate ssDNA binding. Surprisingly, Rad51 interacts with Rad52 at two different bindings sites: one within the positive patch in the disordered C-terminus and the other in the ordered ring. We propose that these features drive Rad51 nucleation onto a single position on the DNA to promote formation of uniform pre-synaptic Rad51 filaments in HR.

Homologous recombination (HR) protects genomic integrity by repairing double-stranded breaks (DSBs)[1–3]. Mutations to genes that encode for HR-related proteins are implicated in a plethora of human cancers and genetic disorders[4,5]. For accurate repair during HR, information from the undamaged sister allele is used as a template. In *Saccharomyces cerevisiae*, HR progresses through a series of defined steps beginning with end resection, where the DSB is processed by the Mre11-Rad50-Xrs2 (MRX) complex yielding 3′ ssDNA overhangs[1]. The transiently exposed ssDNA is coated by the high-affinity ssDNA binding protein Replication Protein A (RPA)[6,7] which then recruits the Mec1-Ddc2 DNA-damage-sensing kinase to regulate downstream events in HR[8,9]. Rad51, the recombinase, catalyzes identification and pairing of the homologous sequence in the undamaged allele. Rad51 forms a helical filament on the ssDNA overhang with defined ATP-driven nucleation and filament growth processes[10,11]. Formation of the Rad51 filament on RPA-coated ssDNA occurs during the pre-synaptic phase of HR. Since Rad51 binds to ssDNA with lower affinity ($K_D \sim 10^{-6}$ M)[12] compared to RPA ($K_D < 10^{-10}$ M)[13,14], mediator proteins are proposed to help overcome the thermodynamic barrier to facilitate ssDNA handoff from RPA to Rad51. Pro-HR mediators function to promote formation of Rad51 filaments whereas anti-HR mediators facilitate Rad51 disassembly[10]. In *S. cerevisiae*, Rad52 is an important pro-HR mediator and *Δrad52* strains are defective in various recombination processes including Rad51-dependent allelic recombination and Rad51-independent single-strand annealing[3,15–19].

While the pro-HR roles of Rad52 are well established, knowledge of the precise mechanism of action remains poorly understood. For example, the canonical model that posits Rad52 promoted RPA-Rad51

[1]Department of Biochemistry and Molecular Biology, Saint Louis University School of Medicine, St. Louis, MO, USA. [2]Department of Chemistry and Biochemistry, Montana State University, Bozeman, MT, USA. [3]Center for Cellular Imaging, Washington University in St. Louis School of Medicine, St. Louis, MO, USA. [4]Department of Biological Sciences, Marquette University, Milwaukee, WI, USA. [5]Present address: Aera Therapeutics, Boston, MA, USA. [6]These authors contributed equally: Jaigeeth Deveryshetty, Rahul Chadda. ✉e-mail: edwin.antony@health.slu.edu

handoff is not consistent with DNA curtain-based single molecule experiments[20,21]. Long ssDNA molecules coated with RPA are readily displaced by Rad51 without the need for Rad52[21]. In similar experiments, when fluorescently-labeled Rad52 molecules are tracked, formation of RPA-Rad52 and RPA-Rad52-Rad51 complexes are observed[21]. Thus, how Rad52 serves as a pro-HR mediator remains to be established. Another tier of complexity arises from the intrinsic structural properties of Rad52. *S. cerevisiae* Rad52 is widely considered to function as a homoheptamer based on negative stain electron microscopy[22] and a single-concentration analytical ultracentrifugation experiment[23]. Each Rad52 subunit can be further divided into an ordered N-terminal and a disordered C-terminal half (Fig. 1A). The primary DNA binding activity and the oligomerization regions are harbored in the N-terminal half[24]. The Rad51 and RPA interaction motifs have been mapped to the disordered C-terminal half of Rad52[25-30]. A weaker DNA binding site has also been identified in the C-terminal half[31,32]. While a high-resolution structure of yeast Rad52 has not been solved, structures of human RAD52 in the apo and DNA-bound forms have provided key mechanistic insights. While both human and yeast Rad52 proteins were initially considered to function as heptameric rings[22,23,33,34], crystal structures of the C-terminal truncated human RAD52 show that the complex is an undecamer[35,36]. Recent cryo-EM analysis of full-length human Rad52 also reveal an undecamer[37]. Since the C-terminal half containing the protein-protein interaction regions are disordered, how the two halves communicate to promote Rad51 filament formation is not understood. It should be noted that human RAD52 is not considered to function as a mediator in HR but evolved to promote strand annealing reactions[38].

Both DNA and protein-protein interactions are integral to the recombinogenic functions of Rad52. When promoting Rad51 filament formation, Rad52 must bind to ssDNA and substrates with a 3′ overhang. During strand annealing, Rad52 must simultaneously bind to two strands of DNA. A new function for Rad52 in RNA-templated DNA repair was recently uncovered[39,40] that would require simultaneous binding to both DNA and RNA fragments[41]. Two crystal structures of the N-terminal half of human RAD52 (residues 1–212) bound to ssDNA have been solved[42]. An outer and inner DNA binding site in the ordered N-terminal half coordinates DNA binding in RAD52[34,43,44]. These sites are well conserved in yeast Rad52 as well. Physical interactions of Rad52 with Rad51 and RPA are essential for the mediator functions in HR, and both binding sites reside in the disordered C-terminal half of Rad52[25-30]. The C-terminus also promotes phase separation of Rad52 into liquid droplets and cluster DNA damage sites[45]. In addition to the above mentioned roles, Rad52 regulates the extent of DNA resection during HR by regulating the Rqh1 helicase in fission yeast[46]. Furthermore, Rad52 functions in other processes such as regulating retrotransposition[47], CRISPR-Cas9 genome editing[48], and IgD class-switch recombination[49]. Given the emerging multifaceted functions of Rad52, knowledge of it functions and mechanism of action is critical.

Towards addressing the missing details about the structure and mechanism of action, using cryo-EM, we here show that *S. cerevisiae* Rad52 functions as a homodecamer with each subunit possessing an ordered N-terminal and disordered C-terminal half. Using cross-linking and hydrogen-deuterium exchange mass spectrometry we capture interactions between the two halves and reveal DNA binding induced conformational changes in both regions. Interestingly, these

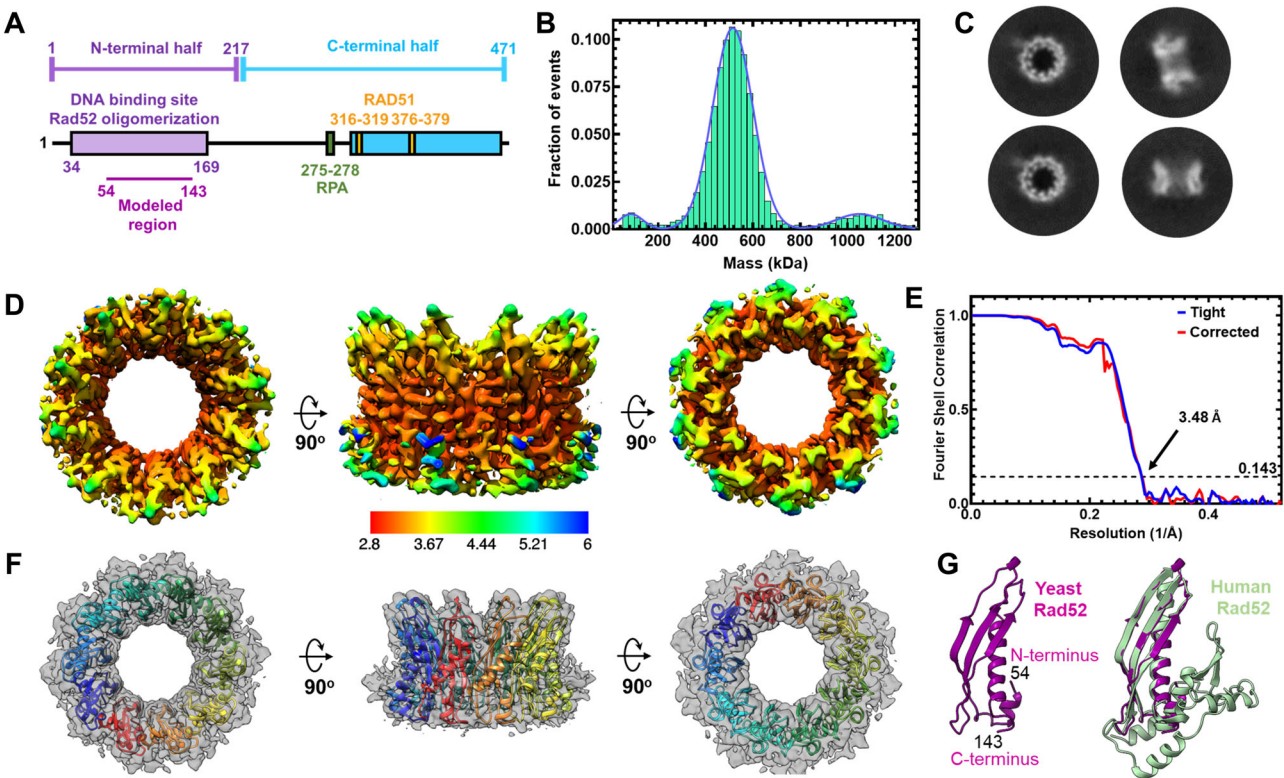

**Fig. 1 | *S. cerevisiae* Rad52 functions as a homodecamer. A** Schematic representation of a yeast Rad52 subunit depicts N- and C-terminal regions that promote DNA binding, oligomerization, & protein–protein interactions. **B** Mass photometry analysis predominantly shows a single Rad52 species. The major species (>85%) shows a mass of 498.4 ± 23.8 kDa. A minor 991.8 ± 75.34 kDa species (~8–12%) is observed along with a small fraction of free monomers (72.9 ± 13.9 kDa). **C** Representative 2D classes of full-length yeast Rad52 observed in cryo-EM analysis and used for determination of the structure. **D** Color coded depiction of the local resolution mapped on to the Rad52 cryo-EM map. **E** Gold Standard Fourier shell correlation (FSC) curves estimated using Cryosparc. **F** De novo built atomic model of Rad52 overlaid on the cryo-EM map. **G** Atomic model of a Rad52 subunit (magenta) aligned with the crystal structure of human RAD52 (gray; PDB 1KN0) using Chimera with root-mean-square deviation (RMSD) of 0.849 Å for Cα atoms of 88 amino acid pairs. Source data are provided as a Source data file.

interactions are not uniform across all Rad52 subunits in the homo-decamer. Only a small subset of the C-termini interacts with the N-terminal ring and dictates ssDNA binding, wrapping, and diffusion properties of Rad52. Furthermore, Rad51 interacts asymmetrically with Rad52 using two binding modes. The first is interactions with the dis-ordered C-terminal half and the other through a well-defined interac-tion with the N-terminal ordered ring. We propose that these asymmetric interactions promote Rad51 nucleation at a single position during HR. Our findings reveal how the ordered decameric N-terminal ring and the disordered C-terminal region cooperate and enable Rad52 to function as a HR mediator to facilitate Rad51 nucleation during HR.

## Results

### *Saccharomyces cerevisiae* Rad52 is a homodecamer in solution

Yeast and human Rad52 are canonically considered to function as heptamers based on negative-stain transmission electron microscopy and analytical ultracentrifugation (AUC) analysis[22,23,33,34]. However, crystal structures of the N-terminal half of human Rad52 revealed an undecamer both in the absence or presence of ssDNA, with ssDNA encircling the ring in a uniform manner[35,36,42]. Thus, to establish the stoichiometry, we performed single-particle cryo-EM analysis of full-length *S. cerevisiae* Rad52. Recombinantly overproduced and purified Rad52 is predominantly a single species in both mass photometry (MP) and analytical ultracentrifugation sedimentation velocity (AUC$^{SV}$) analyses over a range of protein concentrations tested (Fig. 1B and Supplementary Fig. 1). A small fraction (~5–10%) of a higher order species is observed and corresponds to double the mass of the major peak. The predicted mass of one Rad52 subunit is 52.4 kDa and mea-surements from AUC$^{SV}$ yield molecular weights of ~530 kDa that cor-respond to a homodecamer (Supplementary Fig. 1B, C). Mass measurements from MP analysis for such larger disordered proteins show a higher degree of error (Supplementary Table 1).

### Cryo-EM analysis of full-length *S. cerevisiae* Rad52 reveals a homodecamer

2D classification of Rad52 single particles in cryo-EM analysis con-firmed that yeast Rad52 is predominantly a homodecameric ring (Fig. 1C and Supplementary Fig. 2). Rings with 9 or 11 subunits were also rarely observed (<0.2% of all particles). 3D reconstruction and sub-sequent refinement revealed strong density for the N-terminal half (residues 54–143; Supplementary Fig. 3). However, only partial density was observed for the C-terminal half, which is consistent with the disordered properties. The final cryo-EM map reconstruction reached an average resolution of 3.5 Å with local resolution ranging from 2.9–6 Å (Fig. 1D–F and Supplementary Table 2). Using the refined map, a de novo model was built for the N-terminal region encompassing amino acids 54–143. This region of yeast Rad52 is structurally similar to human RAD52 and shares 41% sequence identity[36] with a RMSD of 0.849 Å for Cα atoms of 88 amino acid pairs (Fig. 1G and Supple-mentary Fig. 4). However, there are two key differences. The first is an absence of resolvable density for the petal-like extensions observed around the periphery of the ring structure of human RAD52 (residues 160–208) suggesting they are dynamic in yeast Rad52 (Fig. 1G)[35,36,42]. The second is the presence of extensive positively charged regions on the outer perimeter of yeast Rad52 (Supplementary Fig. 4C). These differences could potentially influence ssDNA and protein–protein interactions in yeast Rad52.

### Cross-linking mass spectrometry reveals interactions between the ordered N-terminal and disordered C-terminal halves of Rad52

The C-terminal half of yeast Rad52 is disordered in our structure. However, this region has been shown to interact with DNA, RPA, and Rad51[32]. Thus, to gain a better understanding of how the ordered N-terminal region and the disordered C-terminal region of yeast Rad52

function together, we performed cross-linking mass-spectrometry (XL-MS) with a bis(sulfosuccinimidyl)suberate (BS3) cross-linker[50]. The *N*-hydroxysulfosuccinimide (NHS) esters positioned at either end of BS3 are spaced 8-atoms apart and report on cross-links with primary amines in Rad52 that are within ~12 Å[51]. Substantial cross-links are observed within the N-terminal ordered region and C-terminal region, along with multiple contacts between the two regions (Fig. 2A). How-ever, upon binding to ssDNA [(dT)$_{97}$] a significant reduction in inter-domain cross-links is observed (Fig. 2B) suggesting that DNA-binding perturbs crosstalk between both domains probably by binding to either and/or both regions followed by their restructuring. Since Rad52 is a homodecamer, we cannot differentiate cross-links that occur within a subunit (*intra*) versus between two distinct subunits (*inter*). For clarity in discussion, we present the data from the perspective of the two halves of Rad52 and focus on the Cross-links between the two halves. Specifically, Lys-84, Lys-117, and Lys-134 in the N-terminal half are positioned close to a defined region in the C-terminal half (residues 340–380; Fig. 2C). Lys-84 and Lys-117 are close to the outer ssDNA binding site whereas Lys-134 is the inner binding site residue (Fig. 2C). Closer examination of the interaction region in the C-terminal half reveals that it harbors the Rad51 binding site. Mutations in this region perturb Rad51 interactions and give rise to HR defects in yeast[26,52]. Given the abundance of conserved positively charged amino acid residues in this region, we term this the '*positive patch*' (Supplemen-tary Fig. 5).

Comparison of the cross-links between the ssDNA bound and unbound conditions reveals a net loss of interactions (Fig. 2A, B). ~38% and ~19% of the cross-links are unique to the unbound and DNA bound Rad52 conditions, respectively. In particular, several of the cross-links observed in the positive patch are lost. These data suggest that ssDNA binding perturbs crosstalk between the N- and C-terminal halves of Rad52 thereby freeing up the C-terminal region. In addition, sub-stantial intra-C-terminal cross-links are captured, suggesting that interactions within the disordered region might be important for function. We note that a conserved '*negative patch*' resides in the C-terminal half of Rad52 (residues 240–280) that harbors the site for RPA binding (Supplementary Fig. 5)[29]. Significant cross-links are observed between the positive and negative patches. We propose that the network of interactions between the patches and the N-terminal half are likely important for function and may dictate DNA binding and other functional properties of Rad52 (discussed below). Correspond-ingly, mutations in these patches result in loss of function phenotypes[26,29,52]. However, it is possible that these mutations affect more than one function of Rad52.

### Hydrogen-deuterium exchange mass-spectrometry (HDX-MS) reveals conformational changes in the C-terminus of Rad52 upon ssDNA binding

XL-MS analysis is limited to the availability of Lys residues to cross-link with BS3. Thus, to map the overall DNA binding driven conformational changes and to identify other potential DNA binding regions, we per-formed HDX-MS analysis of Rad52 in the absence or presence of ssDNA [(dT)$_{97}$]. Deuterium incorporation was quantified as a function of time for Rad52 and the Rad52-ssDNA complex. Peptide level deuterium uptake was compared between conditions (Supplementary Figs. 6–8). The difference in HDX between Rad52 and the Rad52-(dT)$_{97}$ complex (differential-HDX) were mapped onto an AlphaFold model of the Rad52 monomer and shown as a heatmap (Fig. 3A). Regions colored blue or red depict increased or decreased differential-HDX, respec-tively. The AlphaFold model aligns well with the cryoEM structure of the N-terminal half of Rad52 (Supplementary Fig. 9). Differential-HDX changes are observed in the N-terminal half upon DNA binding, spe-cifically, the inner, outer, and transition DNA binding regions (Fig. 3B–D and Supplementary Figs. 6 and 7). Interestingly, a surprising number of peptides in the disordered C-terminal half also show

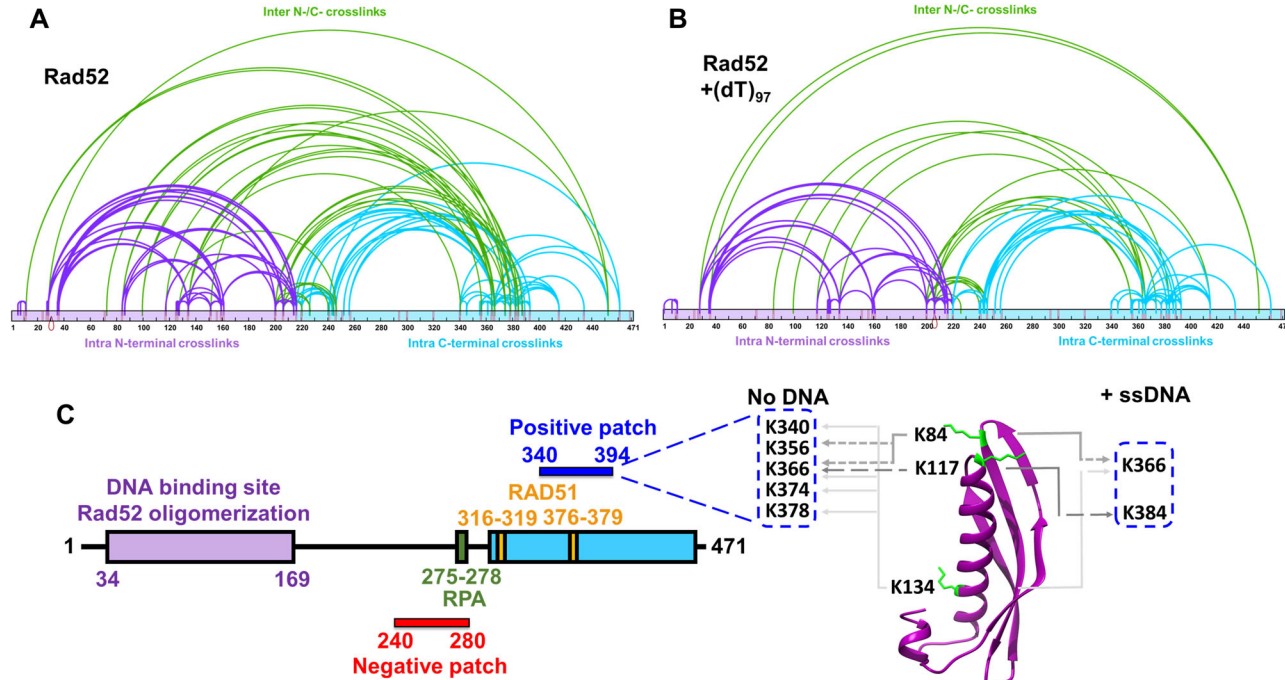

**Fig. 2 | Cross-linking mass-spectrometry (XL-MS) reveals interactions between the N- and C-terminal halves of Rad52. A** Rad52 or **B** the Rad52-(dT)$_{97}$ complex was subject to cross-linking with BS3 and the resulting cross-links were identified using MS. For clarity, data are mapped onto the N-terminal (purple) and C-terminal (cyan) halves of Rad52 as two distinct regions. Intra-N and intra-C half cross-links are colored purple and cyan, respectively. Inter-cross-links between the N- and C-terminal halves are colored green. **C** Schematic representation of Rad52 shows positions of the RPA and Rad51 binding regions. The proposed positive and negative patches are denoted and the key cross-links between the DNA binding sites and the positive patch are highlighted. Source data are provided as a Source data file.

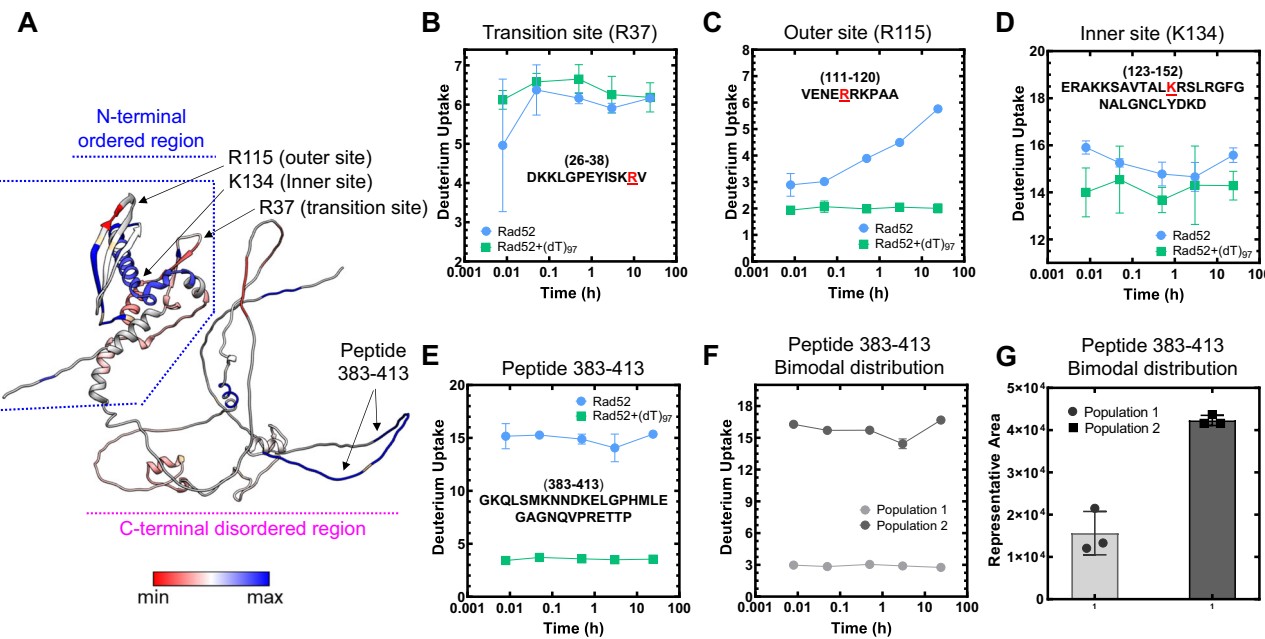

**Fig. 3 | Hydrogen-deuterium exchange mass-spectrometry reveals ssDNA driven changes in both the N- and C-terminal halves of Rad52. A** Differential-HDX changes between Rad52 and the Rad52-(dT)$_{97}$ complex are projected on to the AlphaFold predicted model (AF-P06778-F1). DNA-induced HDX changes in the known DNA binding regions (inner, outer, and transition binding sites) are noted. Raw HDX data showing differences within peptides harboring the **B** Transition (R37), **C** Outer (R115), and **D** Inner (K134) ssDNA binding sites are shown with the key residues highlighted in red. **E** Deuterium uptake profile for peptide 383–413 in the disordered C-terminal half shows a significant reduction in deuterium uptake upon ssDNA binding. **F, G** A bimodal distribution of deuterium uptake was observed in the Rad52 apo sample with least uptake by population 1 and high uptake by population 2. Population 1 represents only 30% of the total signal suggesting that not all subunits in Rad52 are uniform in structure or conformation. Data are representative of at least 3 independent replicates and error bars shown are +/− SEM for each data point. Source data are provided as a Source data file.

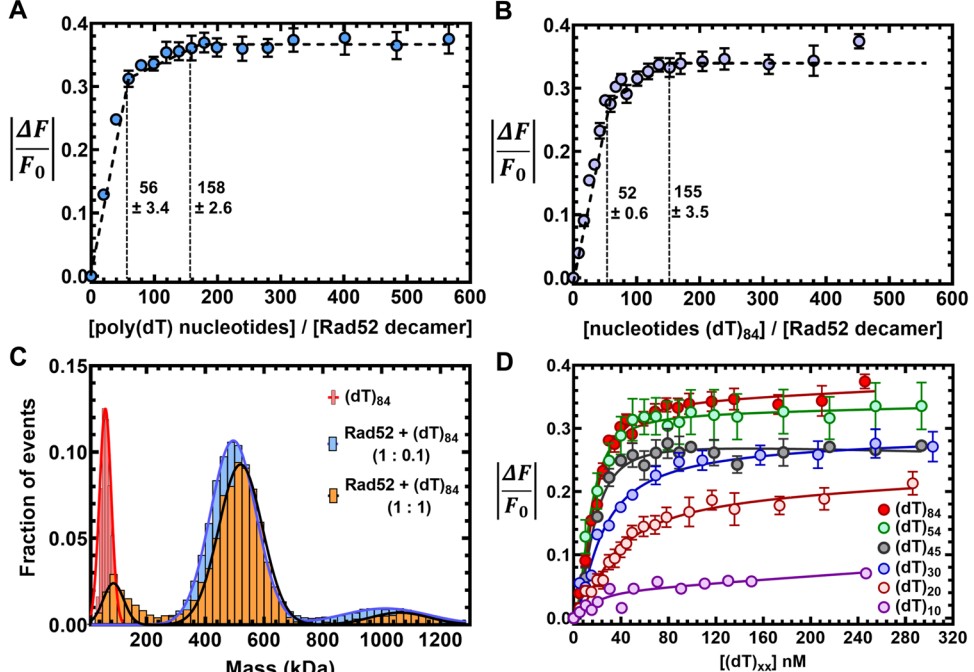

**Fig. 4 | Rad52 binds to ssDNA using multiple binding sites.** Tryptophan quenching experiments with **A** poly(dT) or **B** $(dT)_{84}$ reveal two ssDNA binding phases for Rad52 with sites sizes of $56 \pm 3$ and $158 \pm 3$, respectively. **C** Mass photometry analysis of $(dT)_{84}$ or Rad52 + $(dT)_{84}$ (1:0.1 or 1:1, Rad52$^{decamer}$:ssDNA) show that a single Rad52 homodecamer binds to $(dT)_{84}$. The small fraction of higher order Rad52 molecules also bind to ssDNA and behave as a single DNA-bound complex. **D** Tryptophan quenching of Rad52 as a function of ssDNA length shows that DNA lengths of $>(dT)_{45}$ are required to observe maximal signal quenching. Data are representative of at least 3 independent replicates and error bars shown are +/− SEM for each data point. Source data are provided as a Source data file.

differential-HDX upon ssDNA binding (Fig. 3A, E, F, and Supplementary Fig. 8). One peptide in particular (residues 383–413) displays the largest HDX changes (Fig. 3E). A closer inspection of the MS data for this peptide shows an intriguing bimodal deuterium uptake pattern (Supplementary Fig. 10). In the absence of DNA, two populations of this peptide are observed that show low or high deuterium uptake (Fig. 3F). Calculation of the distribution based on ion intensity shows that ~70% of this peptide displays higher uptake (Fig. 3G). In the presence of ssDNA, a loss of the higher uptake population occurs. The data suggests an intrinsic asymmetry in the structural organization/accessibility of the C-terminal half (C-tails) of Rad52 where a small subset of the C-tails appear to be protected from deuterium uptake.

Since there is a dichotomy in the behavior of C-tails, we reanalyzed our 2D classes from the cryo-EM dataset to gain further insights into this asymmetry. Several 2D classes show the presence of extra bright density adjacent to one part of the decameric ring (Supplementary Fig. 11A). Unfortunately, detailed structural refinement was not possible since this is a disordered region. Low-resolution 3D-volume calculations show the extra density situated on the outer side of the ring closer to the N-terminal half of one side of the Rad52 decamer (Supplementary Fig. 11B). The data reveal an interesting structural feature where only a small subset of the C-tails interact with the decameric ring of Rad52.

**Complex patterns of ssDNA binding by Rad52**

We observed extensive interactions between the N- and C-terminal halves of Rad52 and ssDNA binding altered the interactions (Figs. 2 and 3). Thus, we set out to comprehensively understand the role of these regions in promoting Rad51-DNA interactions. To investigate Rad52 binding to ssDNA, we followed the change in intrinsic tryptophan (Trp) fluorescence. First, to determine the optimal length of ssDNA needed for these studies, we used poly(dT) (average length ~1100 nt) to determine the occluded site-size of Rad52. This is defined as the number of ssDNA nucleotides required to saturate the DNA binding

induced changes in Trp Fluorescence. Two binding phases are observed with site-sizes saturating at $56 \pm 3$ nt and $158 \pm 3$ nt per Rad52 decamer, respectively (Fig. 4A). A similar binding profile is observed on a uniform length ssDNA [$(dT)_{84}$] oligonucleotide (Fig. 4B). The data are well described by a two-site sequential binding model (Supplementary Fig. 12A) and showcases very strong and stoichiometric Rad52-ssDNA interactions. In the DNA-bound crystal structures of human RAD52[42], depending on the mutations introduced to promote crystallization, an inner binding site was observed where a 40 nt ssDNA oligonucleotide wraps around the undecameric ring but interacts with only 10 of the subunits (~4 nt/subunit; Supplementary Fig. 13A)[42]. An outer binding site was also captured with 6 nt of ssDNA sandwiched between two Rad52 undecameric rings (Supplementary Fig. 13B)[42]. In comparison, our data reveals that for yeast Rad52 - 5.6 nt/subunit is required to saturate the high-affinity binding site.

We considered the possibility that the second binding phase might be due to the accumulation of ssDNA bound Rad52 aggregates as has been proposed for human RAD52[42]. To investigate this possibility, we performed both MP and AUC$^{SV}$ analysis to ascertain the stoichiometry of Rad52 decamers bound to ssDNA. When binding of Rad52 to short $(dT)_{45}$ or long ssDNA $(dT)_{97}$ is assessed, stoichiometric Rad52 decamer:DNA complexes are observed (Supplementary Fig. 12B). The minor fraction of dual-Rad52 species also binds to equimolar amounts of ssDNA. On longer $(dT)_{97}$ or $(dT)_{84}$ ssDNA substrates, predominantly 1:1 complexes are observed even under conditions where a 10-fold excess of the Rad52 ring is present (Fig. 4C). Fluorescence anisotropy analysis reveals that Rad52 binding to ssDNA is highly specific (Supplementary Fig. 12C) and shows hallmarks of thermodynamic equilibrium including saturable and high-affinity binding, and reversibility (Supplementary Fig. 12A, C, D). Furthermore, the affinity for ssDNA interactions is dictated by the length of the ssDNA providing an independent confirmation of the binding site size analysis (Fig. 4D; $(dT)_{54}$ is the minimum length of ssDNA which shows similar binding profile as for $(dT)_{84}$). These data suggest that there are

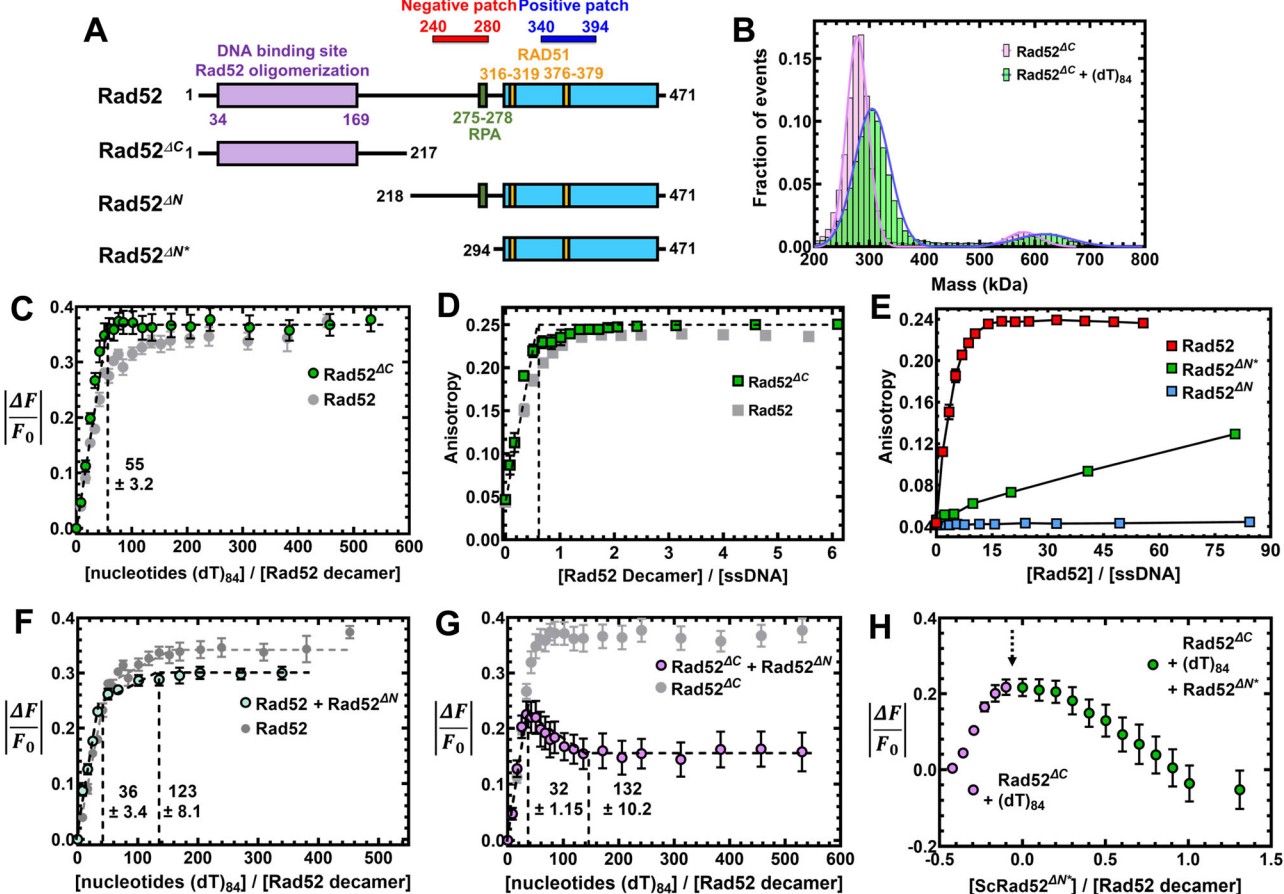

**Fig. 5 | Interactions between the N- and C-terminal halves of Rad52 influence ssDNA binding interactions. A** Schematic of the N- and C-terminal truncations of Rad52. **B** Mass photometry analysis shows Rad52$^{\Delta C}$ as a predominant single species in solution with mass corresponding to a homodecamer (254.7 ± 11.9 kDa). The complex remains a homodecamer when bound to ssDNA. **C** Trp. quenching analysis of Rad52$^{\Delta C}$ binding to poly(dT) reveals high-affinity monophasic ssDNA binding with site-size of 55 ± 3.2 nt/decamer. **D** Fluorescence anisotropy analysis of Rad52$^{\Delta C}$ binding to a FAM-(dT)$_{35}$ oligonucleotide shows stoichiometric high-affinity binding to ssDNA. **E** Fluorescence anisotropy ssDNA binding analysis of N-terminal truncated versions of Rad52 show no ssDNA binding activity for Rad52$^{\Delta N}$, but weak binding for Rad52$^{\Delta N*}$. **F** When Rad52$^{\Delta N}$ is premixed with Rad52, a reduction in the Trp quenching signal upon ssDNA (dT)$_{84}$ binding is observed. The signal corresponds

to the loss of ssDNA binding to the low affinity site in Rad52. **G** When Rad52$^{\Delta C}$ and Rad52$^{\Delta N}$ are premixed and binding to ssDNA (dT)$_{84}$ is assessed through Trp. quenching, an initial increase in binding is observed as the high-affinity site is occupied. This is followed by a sharp loss in binding. **H** When similar experiments are performed by sequential addition, titration of ssDNA (dT)$_{84}$ to Rad52$^{\Delta C}$ produces an increase in Trp quenching as expected (purple data points). After addition of ssDNA, increasing concentrations of Rad52$^{\Delta N*}$ was added, leading to loss in Trp quenching. These data show that defined regions in the C-terminal half of Rad52 modulate ssDNA binding interactions. Data are representative of at least 3 independent replicates and error bars shown are +/– SEM for each data point. Source data are provided as a Source data file.

intrinsic differences in the ssDNA binding properties between the yeast and human Rad52 proteins. Next, we investigated the origins of two-phase ssDNA binding to Rad52.

**The C-tail of Rad52 influences ssDNA binding**

Since the higher order multi-ring properties described for human RAD52 are not applicable to yeast Rad52, we next focused on the contributions of the disordered C-terminal half. We first assessed the ssDNA binding activity of Rad52 mutants carrying deletions in either the C-terminal (Rad52$^{\Delta C}$; residues 1–217) or N-terminal (Rad52$^{\Delta N}$; residues 218–417) halves (Fig. 5A). In MP and AUC$^{SV}$ analysis, majority of Rad52$^{\Delta C}$ behaves as a homodecamer and a smaller fraction (<5%) of higher order species of twice the mass (Fig. 5B and Supplementary Fig. 14A). These properties are similar to full-length Rad52. In cryo-EM analysis, 2D classes reveal that Rad52$^{\Delta C}$ also forms homodecameric rings in agreement with the N-terminal half harboring the oligomerization region (Supplementary Fig. 14D). ssDNA binding analysis reveals that Rad52$^{\Delta C}$ binds ssDNA with similar or higher affinity (Fig. 5C, D), but only the first phase is observed which saturates with a site size of 55 ± 3 nt/decamer (Fig. 5C). This finding strongly suggests that the

relatively weaker second DNA binding site either resides in the C-terminal half or emerges from an inhibited state through protein–protein interactions between the two halves. In addition, each Rad52$^{\Delta C}$ decamer binds to one ssDNA oligonucleotide (Fig. 5B and Supplementary Fig. 14B, C).

The C-tail appears to influence ssDNA binding in the context of full-length Rad52 but a Rad52$^{\Delta N}$ construct (the C-tail by itself, residues 218–417) does not display DNA binding activity (Fig. 5E). However, a shorter version of the C-tail (Rad52$^{\Delta N*}$ residues 294–471) was previously shown to interact with ssDNA in band-shift analysis[32]. In agreement, we also observe weak ssDNA binding activity for Rad52$^{\Delta N*}$ in fluorescence anisotropy measurements (Fig. 5E). We reemphasize that Rad52$^{\Delta N}$ possesses both the positive and negative patches whereas the shorter Rad52$^{\Delta N*}$ protein carries only the positive patch (Fig. 5A). Since we detected cross-links between these two patches in the C-terminus (Fig. 2), we propose that these two regions play opposing roles in regulating ssDNA binding. The positive patch may promote interactions with the phosphate backbone of ssDNA. The negative patch likely suppresses DNA binding/remodeling by sequestering away the positive patch.

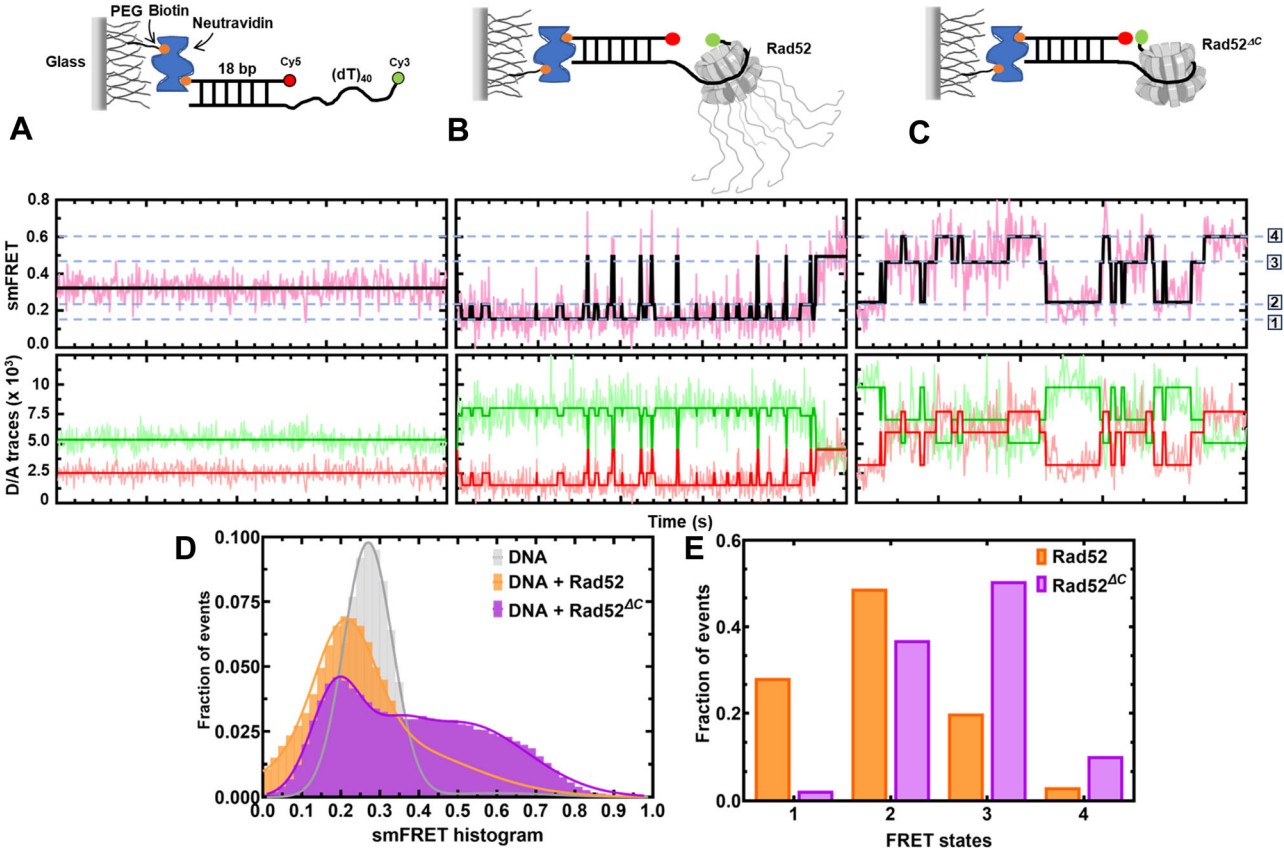

**Fig. 6 | The C-terminus controls intrinsic diffusion of Rad52 on ssDNA.** Single-molecule FRET distributions of **A** (dT)$_{40}$ overhang-ssDNA alone or (dT)$_{40}$ overhang bound to **B** Rad52, or **C** Rad52$^{\Delta C}$. **D, E** HMM (hidden Markov model) analyzed FRET state distributions of smFRET traces. Data show Rad52 preferentially residing in the low FRET states. In contrast, Rad52$^{\Delta C}$ readily accesses and resides in the higher FRET states. The dotted blue lines denote the four observed states. Source data are provided as a Source data file.

We next tested how such interactions might be activated/regulated. First, we tested whether the DNA binding properties of full-length Rad52 are influenced by the presence of free (excess) C-terminal halves in the reaction. Rad52-ssDNA binding measured in the presence of Rad52$^{\Delta N}$ shows a reduction in the second DNA binding phase (Fig. 5F). Furthermore, when Rad52$^{\Delta C}$-ssDNA binding is measured in the presence of Rad52$^{\Delta N}$, an inversion of the Trp quenching profile is observed (Fig. 5G). These data suggest that the second (weaker) binding site likely emerges from the protein–protein interactions between the N- and C-terminal halves. Challenging the ssDNA binding activity of Rad52$^{\Delta C}$ by addition of increasing concentrations of Rad52$^{\Delta N*}$ results in a robust recovery of Trp fluorescence or dequenching (Fig. 5H). In summary, these DNA binding experiments reveal that the ssDNA properties of yeast Rad52 are influenced by a complex non-uniform choreography of interactions between the N- and C-terminal halves. The precise functional roles for such interactions remains to be elucidated.

### The C-tail regulates the dynamic movement/diffusion of Rad52 on DNA

The data shown thus far argues that the two halves of Rad52 harbor distinct ssDNA binding activities: N-terminal half binds a ~56 base long ssDNA tightly followed by a weaker binding to an additional 100 bases either directly to the C-terminal half or to a compound site created by both halves. Furthermore, a small subset of the C-tails interacts with the N-terminal region. For Rad52 to act as an ssDNA annealing machine, the binding of ssDNA must be dynamic. Such 'breathing' of

ssDNA in nucleic acid-protein complexes may facilitate base pairing and annealing. Thus, we hypothesized that the asymmetric interactions between the two halves might regulate the dynamics and diffusion of Rad52 on DNA. Functionally, such regulation could help position Rad52 at a defined position on the DNA (either at the ss-dsDNA junction or at the 3' end). Using single-molecule total internal reflection fluorescence (smTIRF) microscopy, we investigated whether the binding of ssDNA to Rad52 is dynamic. Cy3 and Cy5 fluorophores were positioned on the DNA such that even partial wrapping or unwrapping of ssDNA bound to Rad52 could be captured as distinct FRET states (Fig. 6). A ssDNA length of 45nt is sufficient to capture FRET (Supplementary Fig. 15). The addition of Rad52 to the smFRET DNA substrate led to the appearance of transients i.e., bursts of short-lived, high FRET states (Fig. 6B) which are completely absent in the DNA substrate alone sample (Fig. 6A). This indicates that Rad52 very rarely allows long-lived complete wrapping (high FRET state) of the overhang substrate. This can also be observed in the smFRET histogram (Fig. 6D) with the peak centered at FRET efficiency of 0.2. On the other hand, Rad52$^{\Delta C}$ promotes long-lived high FRET states (Fig. 6C–E and Supplementary Figs. 16 and 17). Taken together, these data strongly indicate that the ssDNA binding/wrapping by Rad52 is highly dynamic with the DNA sampling multiple fully/partially wrapped conformations. Deletion of the C-tail allows the ssDNA to sample higher FRET states more frequently. This indicates that in addition to creating a second (weaker) binding site, the C-tail imposes restrictions on the diffusion of bound ssDNA in the primary (or the high affinity) binding site (Fig. 6B–E).

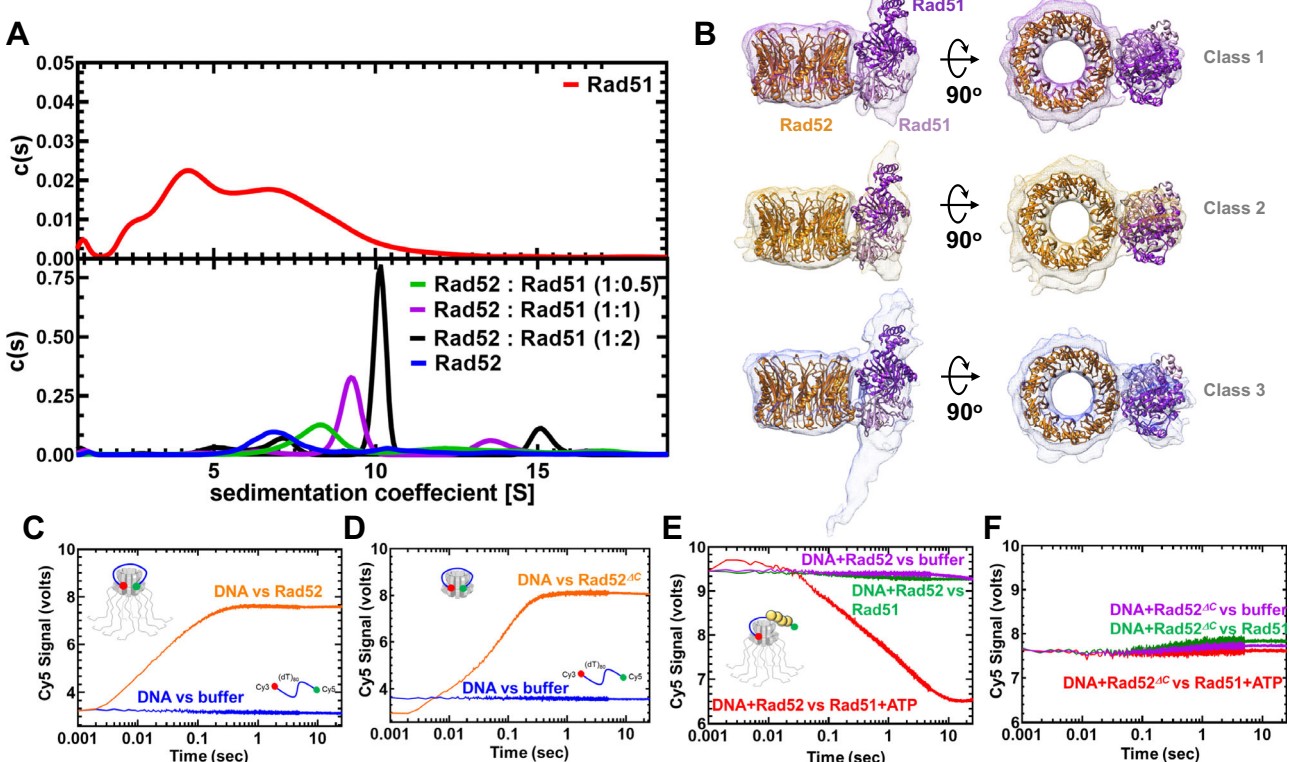

**Fig. 7 | Rad51 interacts with two distinct regions in Rad52. A** AUC$^{SV}$ analysis (top panel) shows polydispersity of Rad51 with multiple oligomeric states. When increasing amounts of Rad51 (1, 10, or 10 μM) are mixed with a fixed concentration of Rad52 (10 μM monomer), formation of Rad52-Rad51 complexes is observed. The size of the complexes increases as a function of Rad51 concentration in the solution. **B** 3D volumes from cryo-EM analysis of Rad52-Rad51 are shown. Structures of Rad52 and Rad51 (1SZP) are fit into the density. 3D volumes generated from three different classes are shown. The data fit well to a dimer of Rad51 bound (ordered) bound to one region in the Rad52 ring. Given the low resolution of the maps, Rad51 can be positioned in multiple orientations in the complex. Higher resolution structures will be required to better understand the details of the interaction. **C** Stopped flow

FRET experiments were performed with a (dT)$_{97}$ ssDNA labeled at the ends with Cy3 (3′; donor) and Cy5 (5′; acceptor). In the absence of protein, no changes in FRET are observed (blue) and a robust increase is captured when Rad52 is introduced (orange). Similar experiments done with **D** Rad52$^{ΔC}$ show wrapping of ssDNA. **E** FRET experiments were performed using prewrapped Rad52-ssDNA complexes and change in FRET was measured upon addition of Rad51 in the absence (green) or presence (red) of ATP. A loss in FRET is observed when Rad51 filaments are formed on the ssDNA in the presence of ATP. **F** Similar experiments performed with Rad52$^{ΔC}$ do not show a decrease in FRET suggesting an impediment to Rad51 filament formation. Source data are provided as a Source data file.

## Rad51 interacts asymmetrically with Rad52 using two distinct binding modes

In both the HDX-MS and cryo-EM analyses, only a select number of C-tails appears to asymmetrically interact with the N-terminus (Fig. 3G and Supplementary Fig. 11). The C-terminal region also harbors the Rad51 and RPA interaction sites. Thus, we next probed whether Rad51 interactions were also asymmetric. Since each Rad52 subunit in the decamer possesses a Rad51 binding region, multiple Rad51 molecules can be expected to be coated around the ring. Furthermore, Rad51 forms multiple oligomeric complexes in solution as observed by AUC$^{SV}$ (Fig. 7A) and MP (Supplementary Fig. 18). When Rad51 and Rad52 were mixed in equimolar amounts (1 Rad51 molecule per Rad52 monomer) all the Rad51 molecules form a complex with the Rad52 decamer (Fig. 7A). Increasing the ratio of Rad51:Rad52 results in formation of complexes with higher molecular masses (Fig. 7A). Thus, multiple subunits in the Rad52 ring are bound to Rad51. To better understand the structure, we analyzed the complex using single-particle cryo-EM (Fig. 7B and Supplementary Fig. 19). Surprisingly, 2D classes show strong ordered density for Rad51 bound alongside only one subunit in the Rad52 decamer (Supplementary Fig. 19A). Even when the concentration of Rad51 is doubled, 2D classes show that the ordered density is observed alongside only one subunit of the Rad52 ring (Supplementary Fig. 19B). Unfortunately, the dynamic nature of the complex hampered our ability to obtain a high-resolution structure. However, analysis of the data yielded three 3D classes where the

density of the Rad52 is clear. The structures of the Rad52 N-terminal ring and a dimer of Rad51 (PDB:1SZP)[53] fit well into the 3D volumes (Fig. 7B).

Two points of contact between the proteins are also visible and Rad51 is situated perpendicular to the axis of the Rad52 ring. This site of interaction is different from the ones described previously in the C-terminus of Rad52[26]. Thus, Rad52 possess bipartite Rad51 binding sites similar to the human BRCA2-RAD51 interactions[54]. However, the density for Rad51 is ordered and observable at a single position on the Rad52 ring. This asymmetry resembles the phenomenon observed between the N- and C-terminal halves of Rad52 (Fig. 3). To test whether the asymmetric C-terminal interactions contribute to the asymmetry in Rad51 binding, we assayed for interactions between Rad51 and Rad52$^{ΔC}$. In this scenario, polydisperse Rad52$^{ΔC}$-Rad51 complexes were observed (Supplementary Fig. 20). Thus, it is likely that the C-tail contributes to implementation of the Rad52-Rad51 binding asymmetry. In addition, the AUC data also support presence of an additional binding site for Rad51 within the N-terminal half of Rad52.

## Formation of proper Rad51 filaments requires the C-tail of Rad52

To investigate the role of Rad52 in promoting Rad51 binding to ssDNA and formation of a filament, we utilized a FRET assay using end-labeled ssDNA oligonucleotides. Cy3 (donor) and Cy5 (acceptor) were positioned at the 3′ and 5′ end of a (dT)$_{97}$ ssDNA oligonucleotide and changes in Cy5 fluorescence were measured by exciting Cy3. Upon

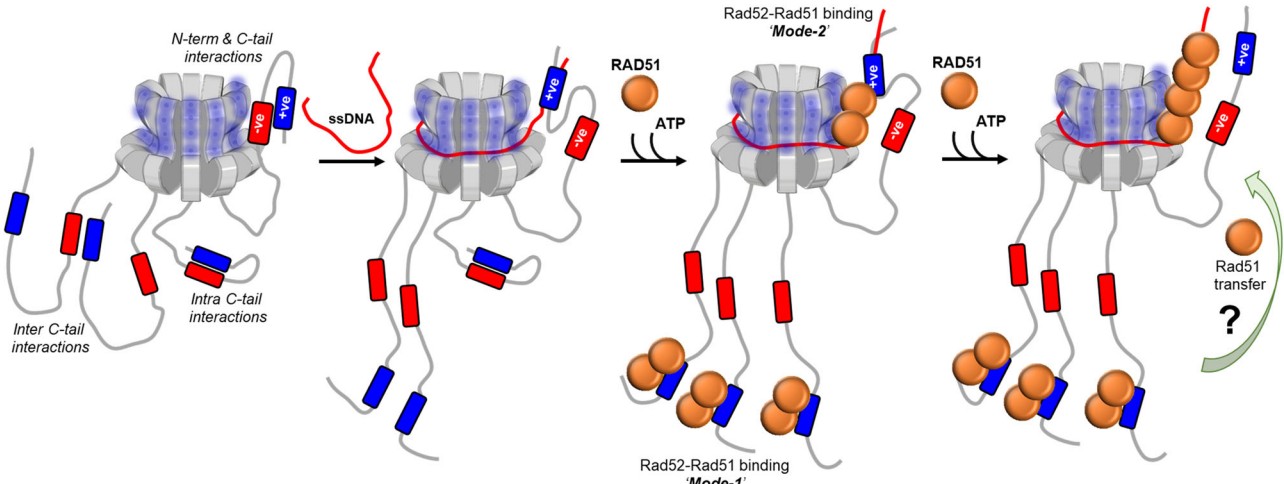

**Fig. 8 | Model describing the mechanism of action of Rad52 and regulation by the C-tail.** The homodecameric ring structure of the N-terminal half of *S. cerevisiae* is depicted and four of the ten disordered C-terminal tails are shown for clarity. The negative and positive patches are shown and the asymmetric interaction between one tail and the N-terminal half is depicted. We propose that such interactions within the remaining tails are suppressed through either intra or inter C-tail interactions between the negative and positive patches. The path of ssDNA binding is modulated through interactions between the DNA backbone and the positive patch bound to the N-terminal half. The bipartite Rad51 interactions are depicted. Mode-1 and Mode-2 are Rad51 interactions with the C- and N-terminal regions, respectively. The binding and remodeling of Rad51 in the presence of DNA and the potential redistribution of Rad51 molecules is speculative. Higher resolution structures will be required to better define the details of the interaction between the two proteins. We propose that the asymmetric interactions promote single-position nucleation and growth of pre-synaptic Rad51 filaments in HR.

binding to Rad52 or Rad52$^{4C}$, the ssDNA is wrapped around the decameric ring, and the fluorophores are brought in proximity resulting in sensitized Cy5 emission (Supplementary Fig. 15 and Fig. 7C, D). To measure the pre-steady state kinetics of this process, experiments were conducted in a stopped flow instrument. Next, preformed Rad52-DNA or Rad52$^{4C}$-DNA complexes were challenged with Rad51 in the absence or presence of ATP. ATP-binding promotes Rad51 binding and filament formation[12], and thus, in the presence of ATP a rapid loss in the FRET signal is observed in experiments with Rad52 (Fig. 7E). In contrast, no reduction in FRET was observed for the Rad52$^{4C}$-DNA complex suggesting that Rad51 does not immediately form a filament when the C-terminal half of Rad52 is absent (Fig. 7F). This finding is consistent with results from single molecule dynamics where a long-lived completely wrapped state with Rad52$^{4C}$ could discourage rapid ssDNA transfer to Rad51 filaments. Interestingly, a minor increase in the FRET signal is observed which reflects Rad51 binding directly to ssDNA as free DNA would be available when Rad52$^{4C}$ is wrapped by a (dT)$_{97}$ substrate (Supplementary Fig. 15). Please note that these changes are only captured in shorter time regimes in the stopped flow experiments. In steady state FRET measurements, Rad51 eventually outcompetes both Rad52 and Rad52$^{4C}$ leading to similar signal changes (Supplementary Fig. 21). Thus, the C-terminus of Rad52 regulates formation of Rad51 filaments on ssDNA.

## Discussion

A barebones view of events during formation of the Rad51 filament during HR pre-synapsis encompasses a resected 3′ ssDNA overhang coated by RPA, Rad51, and Rad52. Previous models suggested that the mediator functions of Rad52 promote removal of the high-affinity DNA binding RPA ($K_D < 10^{-10}$ M) to enable binding/nucleation of the lower affinity Rad51 ($K_D$ - $10^{-6}$ M)[55]. Physical interactions between Rad52 and Rad51 and RPA are required for this exchange[26,30,56]. Recent single-molecule experiments show that RPA-Rad52 function as a complex on the DNA to promote Rad51 binding, with additional roles in downstream second-strand capture[20]. In terms of the structure-function assessments, the N-terminal half of each Rad52 subunit is thought to promote the oligomerization and DNA binding activities[35,36,42,57]. Protein-interaction domains have been

mapped to the C-terminal half [23,26,58,59]. How the various activities of Rad52 are coordinated to enable the mediator functions in HR are poorly resolved.

Using single-particle cryo-EM analysis, we illustrate that *S. cerevisiae* Rad52 functions as a homodecamer. As shown for human Rad52, the N-terminal half is ordered in the homodecameric ring architecture and the C-terminal half is disordered. Remarkably, an intrinsic asymmetry in Rad52 is imparted by the C-terminal region from one (or a select few) subunit that interacts with the ordered ring. This asymmetry in turn regulates a two-state ssDNA binding activity driven through high- and low-affinity ssDNA interactions. When the C-tail is removed, the low-affinity binding is lost. We propose that two charged regions in the C-tail (termed as positive and negative patches) contribute to the asymmetry within Rad52. The positive patch interacts weakly with ssDNA, and interactions between the positive and negative patches inhibit this activity. In the subunits where interactions between the N- and C-terminal halves are observed, the positive charge is accessible to ssDNA interactions. We speculate that such asymmetric interactions could guide the path of ssDNA around the Rad52 ring and likely sets up the polarity and/or define the position of Rad51 or RPA binding (Fig. 8).

In support of this model, the intrinsic asymmetry identified in the Rad52 structure also extends to interactions with Rad51. We find two different binding modes for Rad51: the first is through interactions with the C-terminus of Rad52 and the second via interactions with a single position in the N-terminal ring. This bipartite binding resembles the two classes of BRC repeats in BRCA2 and their differential interactions with RAD51[54]. We propose that the asymmetry in ssDNA and Rad51 binding within Rad52 might serve as a mechanism for 'single-position nucleation' of the Rad51 nucleoprotein filament (Fig. 8). This phenomenon might be essential for accurate DNA repair in HR where identification of perfect homology within the undamaged strand (sister allele) is of paramount importance[60]. For Rad51 (and the prokaryotic homolog RecA) a minimum of 8 nt homology is required to promote strand invasion[61–64]. Furthermore, a 'search entity' has been proposed where Rad51 is in complex with additional proteins (including Rad52 and RPA) to catalyze homology search and pairing[60]. Thus, if Rad51 binds to multiple subunits on Rad52, formation of a

uniform search entity that engages and handles ssDNA at a single position might not be possible. We envision a minimal search entity to contain a Rad52 decamer bound to Rad51 at both positions. Rad52-Rad51 binding through mode-2 adjacent to the N-terminal likely defines the nucleation point for the Rad51 filament on DNA. Whether Rad51 bound on the C-terminus of Rad52 (mode-1) is utilized for filament growth and the underlying mechanisms of Rad51 transfer between the two modes remain to be deciphered.

Another layer of complexity is the position of RPA within the RPA-Rad52-Rad51 complex. Does the minimal search entity comprise RPA-Rad52-Rad51? Does the RPA-Rad52 complex form first and then recruit Rad51, or is it the inverse? Can such complexes be formed in the absence of ssDNA? Many such questions need to be answered. In the absence of ssDNA, we can reliably capture the RPA-Rad52-Rad51 complex in analytical ultracentrifugation analysis (Supplementary Fig. 20C). Using single-molecule tools, we previously showed that Rad52 can selectively remodel DNA-binding domain D of RPA of an RPA-ssDNA complex[13]. In the absence of ssDNA, the Rtt105 chaperone inhibits interactions between Rad52 and RPA by configurationally stapling the flexible architecture of RPA[65]. Thus, many other regulatory mechanisms need to be accounted for to comprehensively understand the nature of the 'search entity' and its regulation.

## Methods

### Reagents and chemicals

Fluorescently labeled and non-labeled oligonucleotides were synthesized by Integrated DNA Technologies Inc. Poly(dT) was purchased from Midland Certified Reagent Company. Mean length of poly(dT) substrate is ~1100 nt. Chemicals were purchased from Millipore-Sigma, Research Products International, Gold Biotechnology, and Fisher Scientific. Resins for protein purification were from Cytiva Life Sciences, Bio-Rad Laboratories, and Gold Biotechnology.

### Plasmids for protein overproduction

Plasmids carrying the coding sequence for yeast Rad52 with a cleavable C-terminal chitin binding domain (pTXB1-CBD) was a kind gift from Dr. Eric Greene (Columbia University). *Rad52$^{ΔC}$, Rad52$^{ΔN}$*, and *Rad52$^{ΔN*}$* were generated by PCR amplifying the respective fragments and subcloning into a pTXB1 plasmid encoding a C-terminal CBD tag using *NdeI* and *XhoI* restriction sites. Plasmids for overproduction of RPA and Rad51 were as described[12,14]. Primers used for generation of Rad52 mutants are listed in Supplementary Table 3.

### Purification of Rad52, Rad52 mutants, and other proteins

Rad52 plasmids were transformed into Rosetta-2 PlysS cells, plated on LB agar plates (with 100 µg/mL ampicillin), and incubated overnight at 37 °C. A single colony was inoculated in a 50 mL LB starter culture supplemented with ampicillin and incubated overnight at 250 rpm at 37 °C. 10 mL of the overnight culture was inoculated into a 2.8 L baffle flask with 1.5 L LB media supplemented with ampicillin and grown at 250 rpm at 37 °C until the growth reached an $OD_{600}$ of 0.6. The culture was induced by the addition of 0.5 mM isopropyl-β-D-1-thiogalactopyranoside (IPTG). Following induction, the culture was incubated overnight at 150 rpm, 18 °C. All the steps from here on were performed at 4 °C or on ice. Cells were pelleted by centrifugation at $5422 \times g$ for 30 min and resuspended in 200 mL cell lysis buffer (50 mM Tris-HCl, pH 8.0, 600 mM NaCl, 1 mM EDTA, and 2X protease inhibitor cocktail (PIC)). Cells were lysed by incubation for 30 min with 0.4 mg/ml lysozyme at 4 °C, followed by 2 cycles of sonication (50% amplitude, 2 s pulses for 60 s) with a minute pause in between. Cell lysate was clarified by centrifugation at $37,157 \times g$ for 1 h. The clarified supernatant was then batch-bound for 3 h on to 20 mL chitin resin (New England Biolabs) equilibrated with 200 mL of cell lysis buffer. The protein-bound beads were sequentially washed with 100 mL each of cell lysis buffer, cell lysis buffer + 1.5 M NaCl and finally with the cell lysis buffer.

Rad52 protein was batch eluted with CBD elution buffer (cell lysis buffer + 50 mM DTT). Chitin column eluates were analyzed by 10% SDS-PAGE and eluates containing Rad52 were pooled, diluted with $R_0$ (50 mM Tris-HCl pH 7.5, 1 mM EDTA pH 8.0, 2 mM βME, 10% glycerol, and 2X PIC) to a final conductivity of $R_{200}$ and applied on a 20 mL heparin column (Cytiva Inc.). Bound Rad52 was fractioned using a linear gradient of $R_{200}$-$R_{1000}$. Fractions containing Rad52 were pooled and loaded on a 20 mL Q-Sepharose column. Fractions containing Rad52 were pooled, concentrated with a 30 kDa cut-off spin concentrator to a volume ~4.5 mL and applied on to HiLoad 16/600 200 Superdex-200 pg size exclusion column. Fractions containing Rad52 were pooled, concentrated with a 30 kDa cut-off spin concentrator, and stored at −80 °C as small aliquots after flash freezing with liquid nitrogen. Before flash freezing, concentration of Rad52 was determined spectroscopically using molar extinction coefficient ($ε_M$) 25,900 $M^{-1}$ $cm^{-1}$. Rad52$^{ΔN}$, and Rad52$^{ΔN*}$ lack tryptophan residues so their concentration was first estimated using the predicted molar extinction coefficient for the peptide bond measured at 214 nm. In addition, the concentration of Rad52, Rad52$^{ΔC}$, Rad52$^{ΔN}$, and Rad52$^{ΔN*}$were also verified using the Bradford method before performing DNA binding experiments.

Rad51 and RPA were purified as described[12,14]. Briefly, *S. cerevisiae* RPA was overproduced in BL21(DE3) Rossetta PlysS cells carrying a plasmid coding for the three RPA subunits. The plasmid expressing all three subunits of RPA (p11d-tscRPA) was a kind gift from Dr Marc Wold (University of Iowa). A C-terminal 6X polyhistidine tag was engineered in RPA32 using Q5 site directed mutagenesis (New England Biolabs, Ipswich, MA). Cells were grown in LB broth supplemented with 50 µg/ml ampicillin at 37 °C to $OD_{600} = 0.6$ and induced with 0.4 mM IPTG overnight at 18 °C. Harvested cells were resuspended in resuspension buffer (30 mM HEPES, pH 7.8, 300 mM KCl, 0.1 mM EDTA, protease inhibitor cocktail, 1 mM PMSF, 10% (v/v) glycerol and 10 mM imidazole). Cells were lysed using 400 µg/ml lysozyme followed by sonication. Clarified lysates were fractionated on a $Ni^{2+}$-NTA agarose column by eluting with 400 mM imidazole. Fractions containing RPA were pooled and diluted three-fold with buffer $Q^0$ (30 mM HEPES, pH 7.8, 0.1 mM EDTA, 1 mM DTT and 10% (v/v) glycerol). The diluted protein sample was then fractionated over a Q-sepharose column equilibrated with buffer $Q^{100}$ (superscript denotes final KCl concentration in the buffer). RPA was eluted with a linear gradient $Q^{100}$–$Q^{400}$. Fractions containing RPA were pooled and diluted with $H^0$ buffer to match the conductivity of buffer $H^{100}$, and further fractionated over a Heparin column. RPA was eluted using a linear gradient $H^{100}$–$H^{1000}$, and fractions containing RPA were pooled and concentrated using an Amicon Ultra spin concentrator (30 kDa molecular weight cut-off). RPA was dialyzed into storage buffer (30 mM HEPES, pH 7.8, 30 mM KCl, 2 mM DTT and 20% (v/v) glycerol), flash frozen using liquid nitrogen, and stored at −80 °C. RPA concentration was measured spectroscopically using $ε_{280} = 98\,500$ $M^{-1}$ $cm^{-1}$.

*S. cerevisiae* Rad51 was overproduced in BL21(DE3) PlysS cells carrying a Plant2b-Rad51 plasmid. Cells were grown in LB broth supplemented with 500 µg/ml Kanamycin at 37 °C to an $OD_{600}$ of 0.6–0.8 and protein overproduction was induced by addition of 0.4 mM IPTG at 37 °C. Cells were harvested after 3 h and suspended in resuspension buffer (100 mM Tris-HCl pH 8.0, 5 mM EDTA pH 8.0, 1 M NaCl, 1 M Urea, 5 mM β-ME, 10% Sucrose, 10% (v/v) Glycerol and protease inhibitor cocktail). Cells were lysed with 0.4 mg/ml lysozyme followed by sonication. The clarified lysate was subjected to ammonium sulfate precipitation (0.24 g/ml) and the pellet was resuspended in buffer $Q^0$ (20 mM Tris-HCl pH 7.5, 1 M Urea, 0.5 mM EDTA pH 8.0, 1 mM β-ME, 10% (v/v) Glycerol). Rad51 was fractionated on a Q-Sepharose column and eluted with $Q^{100}$–$Q^{700}$ (superscript denotes final NaCl concentration in the buffer) gradient. Fractions containing Rad51 were pooled and diluted with $H^0$ buffer (30 mM Tris-HCl pH 7.5, 0.5 mM EDTA pH 8.0, 100 mM NaCl, 0.5 mM β-ME and 10% (v/v) Glycerol) and

fractionated on a Heparin column equilibrated with buffer H[100]. The Heparin column was washed with H[200] and Rad51 was eluted with a linear gradient of H[200]–H[1000]. Heparin fractions containing Rad51 were concentrated to 5 ml volume using an Amicon Ultra spin concentrator (10 kDa molecular weight cut-off). Concentrated Rad51 sample was loaded onto a size exclusion column (Hi Load 16/600 Superdex 200 pg) equilibrated with buffer (20 mM Tris-HCl pH 7.5, 0.5 mM EDTA pH 8.0, 100 mM NaCl, 1 mM β-ME and 10% (v/v) Glycerol). Fractions containing Rad51 were pooled and dialyzed into storage buffer (20 mM Tris-HCl pH 7.5, 0.5 mM EDTA pH 8.0, 100 mM NaCl, 1 mM β-ME, and 20% (v/v) glycerol), flash frozen using liquid nitrogen, and stored at −80 °C. Rad51 concentration was measured using extinction coefficient 11,920 $M^{-1} cm^{-1}$.

## Determination of stoichiometry and molecular weight using mass photometry
Mass photometry measurements were carried out on a TwoMP instrument (Refeyn Inc.). Glass coverslips (No. 1.5H thickness, 24 × 50 mm, VWR) were cleaned by sonication, first in isopropanol, and then in deionized water for 15 min. Cleaned coverslips were dried under a stream of nitrogen. For some measurements (e.g., DNA-bound complexes) coverslips were coated with a 0.1% w/v poly-lysine solution. For each measurement, a clean coverslip with a 6-well silicone gasket was placed onto the water-immersion objective and samples were added onto each well as described below. Molecular weight standards were resuspended in their matching buffers and used to calibrate and convert the particle-image contrast due to scattered light into MW units. Rad52 was diluted to a final concentration of 130 nM decamers (or 1.3 μM monomers) in buffer (50 mM Tris-acetate pH 7.5, 50 mM KCl, and 5 mM $MgCl_2$). After focus was obtained and the image stabilized, 1 μL of the 130 nM protein solution was mixed in with 15 μL of 1X buffer taken in a silicon gasket well resulting in an 8.1 nM solution of decamers (final concentration in the well). Standards included β-amylase (3 species of 56, 112, & 224 kDa, respectively) and thyroglobulin (single species of 667 kDa). 8–10 nM of Rad52 decamers produced statistically significant number of particles (~5000) over a 60 s recording. For data analysis, single particle landings (events) were identified and converted to mass units using the standard calibration, extracted from videos, and non-linear least squares fitted with Gaussian mixture model (Eq. 1) to quantify the underlying populations. For Rad52, the molecular weight upon addition of (dT)$_{84}$ (25.4 kDa) did not appear to shift noticeably. This is likely since the limit of sensitivity of the instrument is ~50 kDa and any increments in molecular weight lower than that cannot be distinguished from noise. However, a change was readily visible for Rad52$^{4C}$.

$$f(x) = \sum_{i=1}^{n} a_i e^{\left[-\left(\frac{x-b_i}{c_i}\right)^2\right]}$$
$$f(x) = a_1 * \exp^{-\left(\frac{x-b_1}{c_1}\right)^2} + a_2 * \exp^{-\left(\frac{x-b_2}{c_2}\right)^2} + \ldots a_n * \exp^{-\left(\frac{x-b_n}{c_n}\right)^2}$$

(1)

Where $a_n$, $b_n$, and $c_n$ are the amplitude, mean, and the standard deviation of the $n$th Gaussian component. For Rad52 and Rad52$^{4C}$, the $n$ equaled 3, but Rad51 showed significant heterogeneity and the Rad51 nano-filament mass size ranged from a dimer to a pentadecamer ($n$ = 15; Supplementary Fig. 18).

## Analytical ultracentrifuge (AUC) analysis
Analytical ultracentrifugation sedimentation velocity experiments were performed with an Optima Analytical Ultracentrifuge (Beckman-Coulter Inc.) using a An-50 Ti rotor. Samples were spun at 40,000 rpm at 20 °C. Samples were prepared by thoroughly dialyzing against 30 mM HEPES, pH 7.8, 100 mM KCl, 10% glycerol, and 1 mM TCEP. Sample (380 μL) and buffer (400 μL) were filled in each sector of 2-sector charcoal quartz cells. Absorbance was monitored at 280 nm for samples using absorbance optics and scans were obtained at 3 min intervals. The density and viscosity of the buffer at 20 °C was calculated using SEDNTERP. The continuous distribution sedimentation coefficient (c(s)) model was used to fit the AUC data using SEDFIT[66].

## Cryo-EM data collection
3 μL of 0.31 mg/mL Rad52 or 0.44 mg/mL of the Rad51-Rad52 complex were applied on to Quantifoil R 2/2 300 copper mesh grids, then plunge-frozen into liquid ethane using a Vitrobot Mark IV (Thermo Fisher Scientific, Brno, Czech Republic) set to 4 °C and 100% humidity. Prior to vitrification, the grids were plasma cleaned in an $H_2O_2$ plasma for 1 min using a Solarus 950 (Gatan, Warrendale, PA). The sample was allowed to incubate for 20 s on the grids prior to blotting. Single-particle cryo-EM data was acquired on a 200 kV Glacios cryo-TEM equipped with a Falcon IV direct electron detector (Thermo Fisher Scientific, Eindhoven, Netherlands). The nominal magnification was 150,000x, resulting in a pixel size of 0.94 Å. The total electron dose was 50.7 e$^-$/Å$^2$ per movie with 48 frames, and the defocus value varied between −0.8 and −2.4 μm.

## Cryo-EM image processing
Single-particle cryo-EM data was processed using cryoSPARC v3.2.0[67]. For the Rad52 dataset, 2447 raw movies were motion corrected using patch motion correction. The CTF estimation of the subsequent micrographs was performed using patch CTF estimation. Initial particle picking was accomplished using blob picker, the results of which were subjected to 2D classification. The best templates were chosen for a subsequent template picker job. After inspection of the template picks, 701,962 particles were extracted using a box size of 300 pixels and subjected to three rounds of 2D classification, resulting in 180,545 particles. Three initial models were generated using an ab-initio reconstruction job, which were then refined by a heterogeneous refinement job with C1 symmetry. After visualizing the volumes in UCSF ChimeraX[68] the best class (consisting of 46,845 particles) was chosen for further processing. The class was subjected to non-uniform refinement with C10 symmetry, then the particle stack was symmetry expanded and underwent local refinement. The resulting cryo-EM density map had an estimated gold standard Fourier shell correlation (FSC) resolution of 3.48 Å at a threshold of 0.143.

For the Rad52-Rad51 complex dataset, 5054 raw movies were motion corrected using patch motion correction, followed by patch CTF job to estimate CTF of the subsequent micrographs. Initial particle picking was accomplished using blob picker, the results of which were subjected to 2D classification. The best templates were chosen for a subsequent topaz job. After inspection of the template picks, 1,111,384 particles were extracted using a box size of 360 pixels and subjected to several rounds of 2D classification and subsequently to several rounds of ab-initio reconstruction and heterogeneous refinement to capture rare views and eliminate non-ideal particles. The final set of 91,315 particles were refined a by heterogeneous refinement job with four input volumes. One of the four classes (44,267 particles) with a well-defined Rad51 density was further refined by heterogeneous refinement with three input classes which revealed extended filament-like density. The heterogeneity in these volumes hampered refinement of these volumes to higher resolution.

## Model building
The Atomic model for yeast Rad52 was built de novo using Model Angelo tool[69]. The Model Angelo generated model was curated manually in Chimera[70] and Coot v0.9.3[71]. The manually curated atomic model was real space refined followed by manual curation in Coot. The final model was validated by Ramachandran plot in Phenix.

## Measurement of ssDNA binding to Rad52 using fluorescence anisotropy

5′-FAM labeled (dT)$_{35}$ or (dT)$_{70}$ ssDNA was diluted to 30 nM in buffer (50 mM Tris-acetate pH 7.5, 50 mM KCl, 5 mM MgCl$_2$, 10% v/v glycerol, 1 mM DTT and 0.2 mg/mL BSA) and taken in a 10 mm pathlength cuvette (Starna Cells Inc). The temperature was maintained at 23 °C as the fluorescein labeled-ssDNA molecules were excited with vertically polarized 488 nm light and emission was collected using a 520 nm bandpass filter in parallel and perpendicular orientation in a PC1 spectrofluorometer (ISS Inc.). The samples were taken in triplicate and five consecutive measurements were averaged for each data point as G-factor corrected anisotropy values were measured after addition of Rad52 (full-length or Rad52$^{ΔC}$). The titration was continued until the anisotropy value plateaued indicating a steady state. The raw anisotropy values were corrected for the reduction in quantum yield of the fluorescein moiety upon Rad52 addition, brought about the proximity-based quenching effects of the bound protein molecule as follows,

$$A_c = \frac{\left[\left(\frac{A-A_f}{A_b-A}\right) \times \left(\frac{Q_f}{Q_b}\right) \times (A_b)\right] + A_f}{1 + \left[\left(\frac{A-A_f}{A_b-A}\right) \times \left(\frac{Q_f}{Q_b}\right)\right]} \tag{2}$$

Where, (1) $F_b$ and $F_f$ are the bound, and free concentrations of the FAM-labeled fluorescent ssDNA in nM, (2) $Q_b$, and $Q_f$ are the fluorescence quantum yields of the bound and free form of the FAM-labeled fluorescent ssDNA (arbitrary units), (3) $A_b$, and $A_f$ are the anisotropy values of the bound and free forms of the FAM-labeled fluorescent ssDNA, (4) $A$, is the measured anisotropy, and (5) $A_c$ is the corrected anisotropy value. Care was taken to limit the maximal dilution to <5% of initial volume and the protein, and ssDNA concentration was corrected for this effect. Due to tight binding between Rad52 (full length, and Rad52$^{ΔC}$), and (dT)$_{35}$, no attempt was made to fit a binding curve to the data points.

## Tryptophan quenching assays

Titrations monitoring the tryptophan fluorescence output of Rad52 (full length or Rad52$^{ΔC}$) were performed with a PC1 spectro-fluorometer. The tryptophan residues in the protein samples were selectively excited with 296 nm excitation light and the emitted fluorescence (320–360 nm) was collected as an emission scan in triplicate using a 10 mm path length cuvette; the peak being located at 345 nm. The temperature of the sample was maintained at 23 °C. ssDNA (poly(dT) or (dT)$_{84}$) was added to a 1.2 mL, 50 nM (Rad52 or Rad52$^{ΔC}$ decamer concentration) protein taken in buffer (50 mM Tris-acetate pH 7.5, 50 mM KCl, 5 mM MgCl$_2$, 10% v/v glycerol, and 1 mM DTT) with constant stirring. After a 2 min incubation, the fluorescence emission scan was reacquired. To minimize photobleaching the excitation shutter was only open during data acquisition. The fluorescence values were corrected for the effects of dilution, photobleaching, and inner filter effects using Eq. 3.

$$F_{i,corr} = F_{i,raw} \times \left(\frac{V_{i,raw}}{V_0}\right) \times \left(\frac{1}{C}\right) \times \left(\frac{f_0}{f_i}\right) \tag{3}$$

Where, $F_{i,raw}$ and $F_{i,corr}$ are measured, and corrected Rad52 fluorescence values after each (ith) addition of nucleic acid stock. $V_0$ is the initial volume of the solution before titration. $V_i$ is the volume after after addition of the ith aliquot. $f_0$, and $f_i$ are the initial fluorescence and fluorescence of Rad52 due to photobleaching alone under identical solution conditions. $C$ or the correction factor for the inner filter effect was assumed to be 1 because of following reasons: (1) The optical density (OD$_{280}$) of Rad52 at 50 nM decamer concentration (or 0.5 μM monomer concentration) is only 0.013 which is at least 7-fold lower than the O.D. = 0.1 cutoff where inner filter effects become significant.

(2) In addition, the normalized fluorescence spectra of Rad52 at 10, 25, 50, and 70 nM decamer concentration appear comparable when normalized for concentration. (3) The maximum concentration of ssDNA in the cuvette, at the end of titration, rarely exceeded 250 nM. The corrected data was plotted as the absolute change in fluorescence normalized to the initial fluorescence ($\left|\frac{\Delta F}{F_0}\right|$) on Y-axis, with [nucleotides]/[Protein] ratio on the X-axis. The biphasic curve resulting from the TRP quenching of Rad52 with poly(dT) or (dT)$_{84}$ could be fitted with a model for two independent, sequential DNA binding sites as follows:

$$TQ = \frac{TQ_1 K_{D1} D_f + TQ_2 K_{D1} K_{D2} D_f^2}{1 + K_{D1} D_f + K_{D1} K_{D2} D_f^2} \tag{4}$$

$$D = D_f + D_b = D_f + \frac{K_{D1} D_f + 2K_{D1} K_{D2} D_f^2}{1 + K_{D1} D_f + K_{D1} K_{D2} D_f^2} [Rad52] \tag{5}$$

Where, $TQ$ is the observed Tryptophan quenching, which is sum of two components $TQ_1$, and $TQ_2$ due to the strong and the weak binding components. $K_{D1}$ and $K_{D2}$ are macroscopic dissociation constants for the first and the second phase, respectively. $D$ is the total DNA concentration which is sum of free and bound components $D_f$, and $D_b$. Note that $D_b$ is a function of total protein concentration or [Rad52]. [Rad52] is assumed to be constant at 50 nM (decamer) during fitting as there is less than 5% dilution by the end of titration. $K_{D1}$, and $K_{D2}$ are determined from the non-linear least squares fitting of Eqs. 2 and 3.

## Hydrogen-deuterium exchange mass spectrometry (HDX-MS) analysis

Stock solutions of Rad52 (15.3 mg/mL) were mixed in the presence or absence of (dT)$_{97}$ ssDNA in a 1:1.2 ratio. Reactions were diluted 1:10 into deuterated reaction buffer (30 mM HEPES, 200 mM KCl, pH 7.8). Control samples were diluted into a non-deuterated reaction buffer. At each time point (0, 0.008, 0.05, 0.5, 3, 30 h), 10 μL of the reaction was removed and quenched by adding 60 μL of 0.75% formic acid (FA, Sigma) and 0.25 mg/mL porcine pepsin (Sigma) at pH 2.5 on ice. Each sample was digested for 2 min with vortexing every 30 s and flash-frozen in liquid nitrogen. Samples were stored in liquid nitrogen until the LC-MS analysis. LC-MS analysis of Rad52 was completed as described[72] Briefly, the LC-MS analysis of Rad52 was performed on a 1290 UPLC series chromatography stack (Agilent Technologies) coupled with a 6538 UHD Accurate-Mass QTOF LC/MS mass spectrometer (Agilent Technologies). Peptides were separated on a reverse phase column (Phenomenex Onyx Monolithic C18 column, 100 × 2 mm) at 1 °C using a flow rate of 500 μL/min under the following conditions: 1.0 min, 5% B; 1.0 to 9.0 min, 5 to 45% B; 9.0 to 11.8 min, 45 to 95% B; 11.8 to 12.0 min, 5% B; solvent A = 0.1% FA (Sigma) in water (Thermo Fisher) and solvent B = 0.1% FA in acetonitrile (Thermo Fisher). Data were acquired at 2 Hz s$^{-1}$ over the scan range 50 to 1700 $m/z$ in the positive mode. Electrospray settings were as follows: the nebulizer set to 3.7 bar, drying gas at 8.0 L/min, drying temperature at 350 °C, and capillary voltage at 3.5 kV. Peptides were identified as previously described[73] using Mass Hunter Qualitative Analysis, version 6.0 (Agilent Technologies), Peptide Analysis Worksheet (ProteoMetrics LLC), and PeptideShaker, version 1.16.42, paired with SearchGUI, version 3.3.16 (CompOmics). Deuterium uptake was determined and manually confirmed using HDExaminer, version 2.5.1 (Sierra Analytics). Heat maps were created using MSTools[74].

## Cross-linking mass spectrometry (XL-MS) analysis

Stock solutions of Rad52 and ssDNA (dT)$_{97}$ were diluted to 1.8 mg/mL and 2.2 mg/mL, respectively, in buffer (30 mM HEPES, 200 mM KCl, pH

7.8) and incubated together for 30 min. The diluted proteins were reacted with primary amine reactive 5 mM bis(sulphosuccinimidyl) suberate (BS3) and 20 μL of the sample was taken at various time points (0, 15, and 30 min) and immediately quenched with 2 μL of 1 M ammonium acetate. Quenched samples were diluted with 1.5X Laemmli gel loading buffer to a final volume of 40 μL, vortexed, and heated to 100 °C for 5 min and resolved on 4–20% (w/v) gradient SDS-PAGE gels (Bio-Rad) with Tris-glycine buffer. Gels were stained with Gelcode blue safe protein stain (Thermo Scientific). Gel bands were excised for protein identification and analysis. Excised bands were destained with a 50 mM ammonium bicarbonate and 50% acetonitrile mixture and reduced with a mixture of 100 mM DTT and 25 mM ammonium bicarbonate for 30 min at 56 °C. The reaction was subsequently exchanged for the alkylation step with 55 mM iodoacetamide and 25 mM ammonium bicarbonate and incubated in the dark at room temperature for 25 min. The solution was then washed with the 50 mM ammonium bicarbonate and 50% acetonitrile mixture. The gel pieces were then first dehydrated with 100% acetonitrile and subsequently rehydrated with sequence grade trypsin solution (0.6 μg, Promega) and incubated overnight at 37 °C. The reaction was quenched with 10 μL of 50% acetonitrile and 0.1% formic acid (FA, Sigma) and transferred to new microfuge tubes, vortexed for 5 min, and centrifuged at $20,627 \times g$ for 30 min. Samples were transferred to mass spectrometry vials and quantitated by LC-MS as described for peptide identification[75,76]. Peptides were identified as previously described[73] using MassHunter Qualitative Analysis, version 6.0 (Agilent Technologies), Peptide Analysis Worksheet (ProteoMetrics LLC), and PeptideShaker, version 1.16.42, paired with SearchGUI, version 3.3.16 (CompOmics). Cross-links were then determined using Spectrum Identification Machine (SIMXL 1.5.5.2).

### Cy3/Cy5 FRET-based steady state ssDNA wrapping experiments

FRET-based analyses of ssDNA binding and wrapping by full-length Rad52 and Rad52$^{\Delta C}$ proteins was carried out by titrating the protein into ssDNA with Cy5 and Cy3 labels at either ends. A 2 mL solution containing 30 nM ssDNA (5′-Cy5-(dT)$_{80}$-Cy3-3′) was taken in 50 mM Tris-acetate pH 7.5, 50 mM KCl, 5 mM MgCl$_2$, 10% v/v glycerol, 1 mM DTT, and 0.2 mg/mL BSA in a 10 mm pathlength quartz cuvette in spectrofluorometer. The temperature was maintained at 23 °C with constant stirring. The samples were taken in triplicate, excited with 535 nm light, and the resulting emission from the donor (Cy3), and the sensitized emission from the acceptor (Cy5) was collected as a FRET scan from 550–700 nm. Ratiometric FRET efficiency was calculated from the background subtracted FRET spectra as follows:

$$FRET_e = \frac{I_A}{I_A + I_D} \qquad (6)$$

Where $FRET_e$ is the FRET efficiency calculated by normalizing the sensitized emission counts ($I_A$; fluorescence intensity collected at 667 nm) with total number of photons collected in the donor and the acceptor channel ($I_A + I_D$; fluorescence intensity counts 667 nm + that collected at 568 nm). We did not estimate or apply correction factors for spectral cross-stalk (i.e., Cy3 emission spilling over into Cy5 channel or vice versa, or cross-excitation (i.e., 535 nm excitation light directly exciting Cy5 fluorophores) because following two reasons: (1) Eq. (6) is a simple means to estimate ratiometric FRET. We are merely interested in monitoring the relative changes in the distance between the ends of the ssDNA upon binding to Rad52 rather than any precise value in nm. (2) A complete battery of correction factors will also include the careful estimation of proximity-based effects of bound protein molecules on the quantum efficiency of Cy3 and Cy5 fluorescence. This will involve additional experiments with Cy3- or Cy5-alone labeled ssDNA of comparable lengths something which is beyond the scope of this present study. After an initial FRET scan of ssDNA alone, full-length

Rad52 or Rad52$^{\Delta C}$ were titrated in multiple steps gradually increasing the protein concentration. After each addition, FRET spectra were reacquired after an incubation period of 2 min allowing the binding reaction to reach equilibrium. The estimated $FRET_e$ was plotted against the concentration of Rad52 in decamer units (see Supplementary Fig. 15) for ssDNA of different lengths. (dT)$_{xx}$, where xx refers to 45, 60, 80, or 97 nucleotides in length.

### Single-molecule analysis of ssDNA-Rad52 wrapping

Single-molecule FRET measurements were performed on an inverted, objective-based total-internal-reflection fluorescence microscope (TIR-FM; IX71 Olympus). TIR excitation was achieved through an oil-immersion objective (100X Olympus UplanApo Numerical Aperture 1.5). The sample was excited with a 532 nm laser and the emitted fluorescence was split into two channels using Optosplit II (Cairn-Research, UK); Cy3 and Cy5 emissions was collected on the half chip of the same electron-multiplying charge-coupled device (EM-CCD) camera (iXon Ultra DU-897U-CS0). Images were acquired with 150 ms frame time. Donor and Acceptor images were subpixel registered in Fiji using an image of multicolor beads as fiducial markers. Next, background corrected intensity traces were extracted from immobilized ingle-molecule spots using Matlab scripts. FRET values were calculated for each fluorescent spot as a ratio between the acceptor intensity and the sum of the intensities of the donor and acceptor, corrected for crosstalk and cross excitation. The single-molecule FRET vs time traces were further analyzed in vbFRET[77] to ascribe FRET states and dwell times.

Glass coverslips for single-molecule imaging were cleaned and coated as follows: Briefly No. 1.5 thickness Gold Seal coverslips (22 ×60 x 0.17) were sonicated in deionized water, 2% Micro-90, 200-proof ethanol, and finally in 1 N KOH all at 60 °C for 15 min each interspersed with exhaustive washes with deionized water. After drying under a stream of filtered N$_2$, the coverslips were first coated with Vectabond (1% v/v Vectabond in 1:18 methanol/acetic acid mixture) followed by washes with deionized water. Next, dried coverslips were incubated overnight with a freshly prepared 18:1 mixture of mPEG-SVA and Biotin-PEG-SVA in 0.1 M NaHCO$_3$. Nonspecific binding, which was measured by flowing in non-biotinylated Cy3- or Cy5-labeled DNA or by flowing in biotinylated fluorescent DNA substrate in absence of Neutravidin, was <1%. All measurements were performed at room temperature in buffer (50 mM Tris-acetate pH 7.5, 50 mM KCl, 5 mM MgCl$_2$, 10% v/v glycerol, 1 mM DTT, and 0.2–0.5 mg/mL BSA) supplemented with oxygen scavenging solution i.e., 0.1 mg/mL glucose oxidase, 0.2 mg/mL catalase, and 0.4% (wt/wt) β-D-glucose (or 0.1 μM PCD/10 mM PCA), and 2.5 mM Trolox to reduce blinking. To a neutravidin (0.2 mg/mL) coated slide, biotin tagged partial duplexes (18 bp duplex region) with 3′ dT overhangs and Cy3/Cy5 pairs (Supplementary Table 3) were added at 50–100 pM concentration to arrive at optimal single-molecule number density of -1.2 spot per 3.5 μm$^2$ area on the EM-CCD chip. Basal FRET (w/o any protein) trajectories were recorded for 5 min. Following which 0.9–1.3 nM (decamer) Rad52 or Rad52$^{\Delta C}$ was introduced. After a 1–2 min incubation period to re-achieve anoxic conditions and the Rad52-ssDNA binding to arrive at an equilibrium, 5 min long videos were again recorded from the same field of view.

### Steady-state analysis of ssDNA handoff to Rad51

To 30 nM ssDNA (5′-Cy5-(dT)$_{80}$-Cy3-3′) taken in buffer (50 mM Tris-acetate pH 7.5, 50 mM KCl, 5 mM MgCl$_2$, 10% v/v glycerol, 1 mM DTT, and 0.2 mg/mL BSA) in a 10 mm path-length cuvette full-length Rad52 or Rad52$^{\Delta C}$ was added until the FRET peaked indicating the end of binding reaction. Next, increasing concentration of Rad51 (±2.5 mM ATP) was added to the reaction in a stepwise manner. As Rad51 concentration was increased, average FRET begins to decrease especially in the case where ATP was present in the reaction. These results are

consistent with the model that ATP-dependent polymerization of Rad51 extracts ssDNA away from Rad52-wrapped configuration and thereby linearizing it and resulting in a loss in FRET.

## Stopped flow analysis of DNA binding and Rad51 filament formation

Stopped-flow experiments were performed using an Applied Photophysics SX20 instrument (Applied Photophysics Inc.) at 25 °C in reaction buffer (50 mM Tris-acetate pH 7.5, 5 mM MgCl$_2$, 10% glycerol, and 1 mM DTT). Reactions from individual syringes were rapidly mixed and fluorescence data were collected. The respective mixing schemes are denoted by cartoon schematics within the figure panel. Five individual shots were averaged for each experiment. All experiments were repeated a minimum of 3 times. In FRET experiments, samples were excited at 535 nm and Cy5 emission was captured using a 645 nm bandpass filter. For the Cy3-dT$_{(80)}$-Cy5 Rad52-Rad51 interactions, Rad52-DNA and Rad52-DNA-Rad51 interactions, experiments were performed with 100 nM each of Rad52 (decamer) and Cy3-dT$_{(80)}$-Cy5 ssDNA substrates, along with 3 μM Rad51. All concentrations denoted are post-mixing.

## Reporting summary

Further information on research design is available in the Nature Portfolio Reporting Summary linked to this article.

## Data availability

The coordinates for the Rad52 structure are available in the PDB under accession code 8G3G and the cryo-EM maps are available in EMBD under accession code EMD-29695. The source data for all experiments shown are provided as a Source data file with this paper. A Chimera session file is provided with the cryo-EM volume and the fitted Rad52-Rad51 structures in the Source data file. Constructs for protein expression are available from the corresponding author upon request. Source data are provided with this paper.

## Code availability

Code used for analysis of single-molecule TIRF data have been deposited in GitHub: https://github.com/ChaddaRah/Single-Molecule-Trajectory-Analyzer_RahulChadda.

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

## Acknowledgements

The authors thank members of our respective research laboratories for critical reading of the manuscript. This work was supported by grants from the National Institutes of Health (R01 GM130746, R01 GM133967, and R35 GM149320) to E.A. Funding for Proteomics, Metabolomics and Mass Spectrometry Facility at MSU was made possible in part by the MJ Murdock Charitable Trust and NIGMS of the National Institutes of Health under Award Number P20 GM103474 to B.B. The analytical ultracentrifugation experiments were supported by an instrumentation grant from the Office of the Director, National Institutes of Health (S10 OD030343 to E.A.). M.J.R. and K.B. gratefully acknowledge support from the Washington University Center for Cellular Imaging (WUCCI) which is funded by Washington University School of Medicine, The Children's Discovery Institute of Washington University and St. Louis Children's Hospital (CDI-CORE-2015-505 and CDI-CORE-2019-813), the Foundation for Barnes-Jewish Hospital (3770) and the Alvin J. Siteman Cancer Center at Washington University School of Medicine and Barnes-Jewish Hospital under NCI Cancer Center Support Grant P30CA091842. J.A.J.F. gratefully acknowledges the Chan Zuckerberg Initiative for their support as a CZI Imaging Scientist under award 2020-225726.

## Author contributions

J.D., M.J.R., K.B. and J.A.J.F. collected single-particle Cryo-EM data and solved the structure. R.C. performed fluorescence and single-molecule experiments. J.M. and B.B. performed and analyzed the HDX-MS and XL-MS experiments. S.K., N.P. and N.E. assisted with cloning, protein purification, and experiments. E.A. conceived the project and wrote the manuscript with assistance and contributions from all authors.

## Competing interests

The authors declare no competing interests.
