## [Peer Review File · Nature Communications]

Yeast Rad52 is a homodecamer and possesses BRCA2-like bipartite Rad51 binding modesREVIEWER COMMENTS

Reviewer #1 (Remarks to the Author):

Rad52 functions in DNA break repair by homologous recombination. In yeast, Rad52 anneals RPA-coated ssDNA, and functions as a recombination mediator to load the strand exchange protein Rad51 on RPA coated ssDNA. In human cells, the mediator function was taken up by BRCA2, and human RAD52 only retained its annealing role. Although we know quite a lot about the general function of Rad52, the mechanisms underlying the mediator and annealing functions are not clear. Rad52 was reported to form oligomeric rings ranging from 10 to 12 subunits. How the ring structure facilitates Rad52 functions is a key question in the field.

The authors here first employ cryoEM and determine the structure of yeast Rad52 as an 11-mer subunit ring with a reasonable resolution. This is a notable achievement. The authors then attempt to determine how Rad52 interacts with DNA and with Rad51 to explain its mediator role. Unfortunately, they were unable to observe in detail complexes of Rad52 with DNA and Rad51 in cryoEM, and instead employ a variety of other approaches that are not always convincing and sometimes quite confusing.

The authors identified interactions between the N- and C-terminal domains of Rad52 that occur only on some of the subunits within the complex. This led to the model that the complex is structurally and functionally asymmetric. This asymmetry is then proposed to drive Rad51 interaction to a single position on the Rad52 oligomer (i.e., one Rad51 molecule per 11-mer Rad52 ring). The latter point, which is the main conclusion of the paper (title), is based on an ultra-low resolution cryoEM data that are only presented in the supplementary material. Instead, in Figure 7, a sedimentation analysis is presented, which concludes, in contrast, that many more Rad51 molecules can interact with the Rad52 ring.

As apparent from the example above that touches upon the main conclusion, I find the conclusions not fully supported by the presented data. The description is often quite dense and difficult to follow.

Specific points

- As noted above, it is unclear why the sedimentation analysis (Fig. 7) and cryoEM (Fig. S19) give different results. The authors should use other methods to determine the stoichiometry. Is it possible that the complexes got disrupted during cryoEM sample preparation? A part of the problem is that the sedimentation coefficient is not only corresponding to mass, but sedimentation velocity may be affected by particle shape.
- The C-term region that interacts with the N-terminal part, as revealed in the crosslinking experiments, is overlapping with the RPA-binding region of Rad52. Are the N-ter-C-ter interactions also detectable in the presence of RPA? As Rad52 loads Rad51 on RPA-coated ssDNA, this is a critical point.
- To verify the importance of the N- and C-ter interactions the authors should design separation of function mutants. The authors note that mutations in the patches that mediate the self-interactions between the N- and C-terminal parts result in loss of function phenotypes. This is however misleading, as the mutations may affect other functions of RAD52 (such as RPA binding). The authors should employ the structure, the crosslinking data and models to design mutants that selectively disrupt the self-interactions, while not affecting the Rad52 ring formation, RPA, Rad51 and DNA binding. They can be then tested to validate the model.
- Have the authors also observed Rad52 open rings, alone or with the protein co-factors?
- Figure 2: I was also puzzled by the XL-MS analysis. Do the crosslinked sites fit with the cryoEM

model of the ring? How can the authors exclude that the observed crosslinks arise from interactions between two adjacent subunits within the ring, or even from interactions between two different rings?

- Figure 4A,B: The text in results wrongly refers to Figure 5

- Figure 5 and 6: The description of the data is difficult to follow, and it is not easy to imagine the second binding site (cartoon?). Could the bi-phasic curve simply result from full-length Rad52 sliding along DNA?

- While the introduction is already quite long, an important information is missing, i.e. that human RAD52 does not function as a mediator. It would be helpful to mention it as both human and yeast proteins are discussed

Reviewer #2 (Remarks to the Author):

Deveryshetty et al describe structural and biochemical studies of the yeast Rad52 recombinational mediator protein. The manuscript is highly multidisciplinary, with solution and cryo-EM structural studies complemented by well-executed DNA and Rad51 binding experiments. The combination of approaches provides new insights into the mechanism used by mediator proteins, with an important emphasis on the yeast Rad52 protein, which has been a topic of research for many decades. The final model of Rad52 presented in the paper is well supported by the reported experimental findings and will serve as an important reference for ongoing studies that aim to understand cellular and biochemical mechanisms of homologous recombination. The paper is very well written and I have on minor criticisms.

(1) Page 5, first results paragraph. Solution studies point to a possible arrangement of yeast Rad52 as a 10-mer but the conclusions are too strongly worded for the data presented. First, the terms "functions" and "functional" should be stricken – the results support a 10-mer arrangement for the protein under the conditions that being examined but these are not "functional" conditions with DNA and other recombinational repair proteins. Whether there is a single state that is functional has not been addressed in the presented experiments. Second, without an error analysis for the MP or AUC, the precision of the measurements are not clear and the extent to which the data rule out arrangements beyond 10-mers is missing. Although the MP mass is provided as 498.4+/-13.9 kDa in the legend for Fig 1, the mass is listed as "~500 kDa" in the results, which does not suggest measurements that allow dismissal of multiple possible oligomeric arrangements. Although the EM structure supports the proposed quaternary structure, the data presented in the first section are not conclusive enough to warrant the strength of the conclusions.

(2) Related to point (1), the title of Supp Fig 12, "Rad52 binds to ssDNA as a homodecamer" is overly definitive with regard to the oligomeric state that Rad52 assumes when bound to ssDNA.

(3) Pages 7-8. It is a bit odd that two very basic regions interact with one another – why would these not repulse given their charges? It may be that this apparent oddity arises from the length of the crosslinker used in XL-MS connecting regions that do not directly interface but instead reside nearby each other. A discussion of this point should be included in the results and/or discussion section.

(4) Fig 3A. I don't understand the color scheme for the HD-MX data. From the results section, I believe that "blue" represents peptides where HDX is increased when dT97 is bound and "red" represents peptides where HDX is decreased when dT97 is bound. But the bar in Fig 3A is listed as "min" and "max" for blue and red. Which is it?

(5) EM map rendering in Supp Fig 3B,C is not very informative. I don't know how these maps were made, but map features are lost in the panels.

(6) Pages 7-8. The XL-MS data are quite rich and informative. Is it clear that the negative patch from one subunit interacts with its own N-terminal domain? This is the impression that one is left with from Fig 2. If it is possible that the C-terminal region interacts with any N-terminal domain within Rad52, this should be pointed out explicitly.

(7) Page 12, line 298-299. The word "it" is missing from the sentence that reads "Although the C-tail appears..."

Reviewer #3 (Remarks to the Author):

Rad52 is an essential protein that promotes homologous recombination by facilitating the formation of Rad51 filaments formation in *S. cerevisiae* by catalyzing the replacement of RPA by Rad51 on ssDNA. To date, the detailed mechanism by which Rad52 promotes Rad51 filament formation has yet to be elucidated. Furthermore, as the C-terminal region of Rad52 is a disordered region, how this region behaves within the Rad52 molecule and affects binding to ssDNA remains enigmatic. This manuscript reports the decameric structure of *S. cerevisiae* Rad52 and provides interesting insights into the regulation of ssDNA binding by Rad52 via its C-terminal region. However, as commented below, the overall results do not support the assertion of this paper that the decameric Rad52 could serve as a mechanism for the "single-position nucleation" of the Rad51 filament. The authors should reconsider their cryo-EM analyses to improve the quality of the structures obtained and to strengthen their conclusion. At least, the discussion needs to be reconsidered and pursued more carefully. The authors are therefore strongly encouraged to respond to the following comments to improve the overall quality of the manuscript.

Major Comments:

Comment 1: Regarding the cryo-EM analysis: due to the symmetry of the decameric structure of the N-terminal ring, it may be that the density corresponding to Rad51 converges to a specific position during the 2D classification process, even though Rad51 binds to multiple positions in the Rad52 ring structure. Consequently, the results of their cryo-EM structural analysis are not sufficient to allow the authors to conclude that Rad52 catalyzes Rad51 binding at a single position.

Comment 2: In the structural analysis of the full-length Rad52-Rad51 complex, the density observed around the decameric Rad52 could be due to the C-terminal region of Rad52. The authors should improve the analysis to a resolution that would enable a structural model of Rad51 to be fitted.

Comment 3: Supplementary Figure 2: The authors are encouraged to improve the quality of the cryo-EM analysis of Rad52. It seems to me that in the graph, especially the upper and middle one, the loop of the C-term of the human full length-Rad52 can be vaguely detected. By increasing the number of particles in the analysis, the authors may be able to improve the 2D averaging step to correctly see the C-term.

Minor comments:

Comment 1: P7, lines 176 and 184: "Fig. 2A and B" is the appropriate citation for the figures in the text.

Comment 2: P10, line 249: "Fig. 4A" is the appropriate citation for the figure in the text.

Comment 3: P10, line 257: " Supplementary Fig. 13B" is the appropriate citation for the figure in the text.

Comment 4: Figure 3B appears to be a modified figure of the graph in the first row and second column of Supplementary Figure 6. However, the error bars have been removed and must be added.

Comment 5: In the schema shown in Figure 6, The DNA to which Cy5 and Cy3 have been added does not correspond to the fluorescent molecules attached to the oligos in Supplementary Table 3, which needs to be corrected.

Comment 6: In Figure 6B, the illustration of the C-terminal region of RAD52 needs to be corrected because it overlaps the graph below.

Comment 7: Figure 6E: the authors must specify what the FRET states (1, 2, 3 and 4) correspond to.

Comment 8: On P.25 in the Materials and Methods section, some characters have been replaced by white squares. Please correct them.

Response to reviewer comments.

We thank the reviewers for their time, thoughtful comments and suggestions. We have addressed all the points raised. Our responses are in blue. The improvement in the Rad51-Rad52 dataset allows us to propose a BRCA2-like bipartite Rad51 binding mechanism. The changes in the title and the manuscript reflect this finding. We thank the reviewers for guiding us down this path.

There is a lot more work to do and new paths have emerged, but making mutations and characterizing these will be beyond the scope of the current manuscript. The updated version of the paper is dense and packed with a lot of new findings that prompt us to rethink how Rad52 functions. We hope the reviewers agree with this assessment and allow us to publish the updated version of the manuscript.

Reviewer #1 (Remarks to the Author)

Rad52 functions in DNA break repair by homologous recombination. In yeast, Rad52 anneals RPA-coated ssDNA, and functions as a recombination mediator to load the strand exchange protein Rad51 on RPA coated ssDNA. In human cells, the mediator function was taken up by BRCA2, and human RAD52 only retained its annealing role. Although we know quite a lot about the general function of Rad52, the mechanisms underlying the mediator and annealing functions are not clear. Rad52 was reported to form oligomeric rings ranging from 10 to 12 subunits. How the ring structure facilitates Rad52 functions is a key question in the field.

The authors here first employ cryoEM and determine the structure of yeast Rad52 as an 11-mer subunit ring with a reasonable resolution. This is a notable achievement. The authors then attempt to determine how Rad52 interacts with DNA and with Rad51 to explain its mediator role. Unfortunately, they were unable to observe in detail complexes of Rad52 with DNA and Rad51 in cryoEM, and instead employ a variety of other approaches that are not always convincing and sometimes quite confusing.

The authors identified interactions between the N- and C-terminal domains of Rad52 that occur only on some of the subunits within the complex. This led to the model that the complex is structurally and functionally asymmetric. This asymmetry is then proposed to drive Rad51 interaction to a single position on the Rad52 oligomer (i.e., one Rad51 molecule per 11-mer Rad52 ring). The latter point, which is the main conclusion of the paper (title), is based on an ultra-low resolution cryoEM data that are only presented in the supplementary material. Instead, in Figure 7, a sedimentation analysis is presented, which concludes, in contrast, that many more Rad51 molecules can interact with the Rad52 ring. As apparent from the example above that touches upon the main conclusion, I find the conclusions not fully supported by the presented data. The description is often quite dense and difficult to follow.

We want to clarify that our structure establishes ScRad52 as a 10-mer. We have collected more cryoEM data and the Rad52-Rad51 complex 3D volumes confirms our model. However, based on the insightful comments, we have also refined our model to explain the AUC results as well (detailed below). We hope our revisions help address the reviewer's concerns.

Specific points

1. As noted above, it is unclear why the sedimentation analysis (Fig. 7) and cryoEM (Fig. S19) give different results. The authors should use other methods to determine the stoichiometry. Is it possible that the complexes got disrupted during cryoEM sample preparation? A part of the problem is that the sedimentation coefficient is not only corresponding to mass, but sedimentation velocity may be affected by particle shape.

We do not consider the AUC and cryoEM data as contradictory results. They each reveal important details about the stoichiometry that will be missed if not considered together. The clear shift in the sedimentation value at 10:1 and 20:1 ratio of Rad51 and the homodecameric ring shows that all the Rad51 molecules in the reaction are indeed bound to Rad52. The cryoEM data are always going to be limited by the ordered part of the complex.

The reviewer is correct in her/his assessment, and we need to consider a model that fits both findings. We considered two possibilities, the first is that the Rad51 molecules are assembling as a filament perpendicular to the axis of the ring with preferential binding to one (or a subset) of the Rad52 subunits. But, such a complex would require ATP (and likely DNA) to form a stable nucleoprotein filament. A more likely scenario is that multiple (or all) C-tails in Rad52 are binding to Rad51 (either a monomer or dimer of Rad51).

Our higher resolution cryoEM data of the Rad52-Rad51 (now shown in Figure 7B) answers this question and helps present a better model. The cryoEM data show that Rad51 molecules bound to one Rad52 subunit are indeed stabilized and better ordered compared to the others. The other tails and their bound Rad51 are not captured in the cryoEM analysis because they are disordered and moving around. The data show two points of contact between the ordered N-terminal ring of Rad52 and Rad51. These are new binding sites that we not previously identified. We have docked (fit) in the crystal structures of Rad51 into the map and the data are in excellent agreement with our model. So now, we must consider bipartite Rad51 binding sites on Rad52: one in the disordered C-terminal half and another in the N-terminal half. How Rad51 binding is partitioned between these binding sites on Rad52 remains to be uncovered. As suggested by the reviewer, and the additional data we have collected and analyzed, we moved the Rad52-Rad51 cryoEM data to the main Figures. We also edited our model to incorporate these alternate possibilities and clearly define the unknowns using '?' marks.

Based on the suggestions and the new data we present a BRCA2-like Rad51 binding mechanism. In BRCA2, there are two functional classes of BRC repeats that differentially interact with RAD51. Thus, we propose that the N- and C-terminal interacting Rad51 molecules might be used differently by Rad52.

2. The C-term region that interacts with the N-terminal part, as revealed in the crosslinking experiments, is overlapping with the RPA-binding region of Rad52. Are the N-ter-C-ter interactions also detectable in the presence of RPA? As Rad52 loads Rad51 on RPA-coated ssDNA, this is a critical point.

While this is a good suggestion, there is a technical issue in performing and interpreting the experiment with RPA. As part of another ongoing cryoEM study, we are investigating the Rad52-RPA complex in the absence or presence of DNA (+/- Rad51). While not ready for publication, AUC, MP, and preliminary cryoEM data show that only one RPA molecule interacts with the Rad52 ring. Similar to the Rad52-Rad51 complex, the asymmetry in interactions also extend to RPA-Rad52 binding. Thus, in XL-MS experiments, most of the C-tails will be unbound while others might be bound to RPA. Thus, we will not form an uniform complex (in terms of the C-tails of Rad52) and the crosslinks will be a mixture of the two species. Thus, we will not be able to tease apart the details requested.

An HDX-MS analysis of the RPA-Rad52 complex +/-DNA might be a better experiment but is beyond the scope of this study. We will be carrying out these experiments as part of the Rad52-RPA-Rad51 story in the future.

3. To verify the importance of the N- and C-ter interactions the authors should design separation of function mutants. The authors note that mutations in the patches that mediate the self-interactions between the N- and C-terminal parts result in loss of function phenotypes. This is however misleading, as the mutations may affect other functions of RAD52 (such as RPA binding). The authors should employ the structure, the crosslinking data and models to design mutants that selectively disrupt the self-interactions, while not affecting the Rad52 ring formation, RPA, Rad51 and DNA binding. They can be then tested to validate the model.

I'm sure the reviewer would concur that nothing about yeast Rad52 is straightforward based on our data. Every assumption that we made based on models in the literature has been incomplete or incorrect. This applies to Rad52-DNA binding, Rad52-Rad51 and Rad52-RPA interactions. While our manuscript here details many interesting features and findings, how they all fit together needs to be worked out.

Addressing this point about making separation of function mutants as an addition to the data presented here is too much to add to this already dense manuscript. At this point, we are not even certain whether the DNA binding contacts described in the literature are correct for Rad52. For example, making outer and inner binding site mutations should be straightforward and result in a loss of the high affinity DNA binding activity. We observe that this not the case.

We are talking about a region that is over 150 aa in length and narrowing down a separation of function mutant when threading the needle between DNA binding, RPA binding, Rad51 binding, and intra-Rad52 interactions is not a trivial amount of work and warrants a study on its own.

The statement we make in the paper denotes that mutations have been explored in this region and compromises overall HR activity in cells. We modified the statement in the paper to further address the reviewer's concern: *"We propose that the network of interactions between the patches and the N-terminal half are likely important for function and may dictate the DNA binding and other functional properties of Rad52 (discussed below). Correspondingly, mutations in these patches result in loss of function phenotypes.*

However, it is possible that these mutations affect more than one functional property of Rad52”.

4. Have the authors also observed Rad52 open rings, alone or with the protein co-factors?

We presume this question is inspired by new unpublished findings on the hRAD52 protein from the West group. In our experiments, more than 99% of molecules in the 2D classes are closed rings. The protein is a 10-mer when bound to RPA, Rad51, ssDNA or when in complex with all components. We also have new data on 3' ssDNA overhangs with both RPA and Rad51 and in all these cases we observe Rad52 functioning as a closed-ring decamer.

5. Figure 2: I was also puzzled by the XL-MS analysis. Do the crosslinked sites fit with the cryoEM model of the ring? How can the authors exclude that the observed crosslinks arise from interactions between two adjacent subunits within the ring, or even from interactions between two different rings?

In the XL-MS analysis one cannot differentiate interactions between the subunits versus interaction within a single subunit. This is why we hypothesize that both inter- and intra- subunit interactions are possible as noted in our model. The second possibility of two rings interacting should be extensively suppressed because a majority (>90%) of Rad52 behaves as a single species in solution. We do not exclude any of these possibilities and as suggested by the reviewer, the inter-subunit interactions are proposed in the model.

6. Figure 4A,B: The text in results wrongly refers to Figure 5.

Corrected.

7. Figure 5 and 6: The description of the data is difficult to follow, and it is not easy to imagine the second binding site (cartoon?). Could the bi-phasic curve simply result from full-length Rad52 sliding along DNA?

We agree on the complexity of the DNA binding properties. These were quite surprising to us and have made our cryoEM efforts refractory to obtaining a structure of the Rad52-ssDNA complex. Even attempts with DNA of multiple lengths yield very dynamic complexes. This behavior is drastically different from human RAD52 that readily produces ssDNA-wrapped rings.

We refrained from drawing a cartoon because there are multiple possibilities that we cannot readily sort out. One is a model where DNA binding between the inner, outer, and C-terminal regions is non-uniform. The second possibility is that such interactions are looping the ssDNA between subunits. Again, this is not very straightforward (or uniform) as mutations in these regions do not help ascertain their functional contributions.

The binding site in the C-tail is likely not a major contributing factor for the first binding phase because the high affinity interaction is not affected in the Rad52^{ΔC} protein. These are bulk experiments and sliding cannot be singled out as a reason for the multiple binding phases, but sure is an integral component of the DNA binding mechanism.

We have extensively reworked this section and hope it reads better. We hope to resolve many of the questions down the road as we investigate the complexities of DNA interactions.

8. While the introduction is already quite long, an important information is missing, i.e. that human RAD52 does not function as a mediator. It would be helpful to mention it as both human and yeast proteins are discussed.

We included this statement in the introduction. *“It should be noted that human RAD52 is not considered to function as a mediator in HR, but evolved to promote strand annealing reactions.”*

Reviewer #2 (Remarks to the Author)

Deveryshetty et al describe structural and biochemical studies of the yeast Rad52 recombinational mediator protein. The manuscript is highly multidisciplinary, with solution and cryo-EM structural studies complemented by well-executed DNA and Rad51 binding experiments. The combination of approaches provides new insights into the mechanism used by mediator proteins, with an important emphasis on the yeast Rad52 protein, which has been a topic of research for many decades. The final model of Rad52 presented in the paper is well supported by the reported experimental findings and will serve as an important reference for ongoing studies that aim to understand cellular and biochemical mechanisms of homologous recombination. The paper is very well written, and I have on minor criticisms.

Thank you for the supportive assessment of our work.

(1) Page 5, first results paragraph. Solution studies point to a possible arrangement of yeast Rad52 as a 10-mer but the conclusions are too strongly worded for the data presented. First, the terms “functions” and “functional” should be stricken – the results support a 10-mer arrangement for the protein under the conditions that being examined but these are not “functional” conditions with DNA and other recombinational repair proteins. Whether there is a single state that is functional has not been addressed in the presented experiments. Second, without an error analysis for the MP or AUC, the precision of the measurements are not clear and the extent to which the data rule out arrangements beyond 10-mers is missing. Although the MP mass is provided as 498.4±13.9 kDa in the legend for Fig 1, the mass is listed as “~500 kDa” in the results, which does not suggest measurements that allow dismissal of multiple possible oligomeric arrangements. Although the EM structure supports the proposed quaternary structure, the data presented in the first section are not conclusive enough to warrant the strength of the conclusions.

We removed both terms as suggested. However, in addition to the MP and AUC data, we are certainly privy to additional unpublished Rad52 structural and biochemical data that show Rad52 as a homodecamer bound to ssDNA, RPA, Rad51, RPA and Rad51, and Rad51 filaments in the presence of nucleotide and on 3' overhangs (please see image below).

(2) Related to point (1), the title of Supp Fig 12, “Rad52 binds to ssDNA as a homodecamer” is overly definitive with regard to the oligomeric state that Rad52 assumes when bound to ssDNA.

As mentioned above, we are working towards a ssDNA bound structure, but it has been difficult to say the least. We did not want to add another set of unresolved 3D volumes to the manuscript. But, to the reviewer’s point, shown above are the 2D classes for Rad52 on three different DNA substrates. It is always a decamer.

(3) Pages 7-8. It is a bit odd that two very basic regions interact with one another – why would these not repulse given their charges? It may be that this apparent oddity arises from the length of the crosslinker used in XL-MS connecting regions that do not directly interface but instead reside nearby each other. A discussion of this point should be included in the results and/or discussion section.

The stretch of negative residues in the ‘negative patch’ is pretty clear (please see electrostatic region in image). While it should be taken with a grain of salt, AlphaFold predicts this region to bind the DNA binding groove. This likely explains the second phase in the DNA binding data as well. When this region is removed,

the second DNA binding phase is lost. There are four Lys residues (K241, K246, K21 and K257) that precede this region.

The 'positive patch' is rich with Lys and Arg residues. The availability of Lys residues adjacent to the Negative patch and many in the Positive patch are likely involved in the crosslinks observed in the XL-MS analysis with BS3. While these experiments are limited by distance of the crosslinks, without structural evidence, it remains a speculative aspect in our model.

As suggested, we reworked this section to better define these interactions and the limitations of XL-MS.

(4) Fig 3A. I don't understand the color scheme for the HD-MX data. From the results section, I believe that "blue" represents peptides where HDX is increased when dT97 is bound and "red" represents peptides where HDX is decreased when dT97 is bound. But the bar in Fig 3A is listed as "min" and "max" for blue and red. Which is it?

The heat map insert was incorrectly labeled. Apologies. This has been corrected.

(5) EM map rendering in Supp Fig 3B,C is not very informative. I don't know how these maps were made, but map features are lost in the panels.

This Figure has been remade. We hope the map is clearly visible.

(6) Pages 7-8. The XL-MS data are quite rich and informative. Is it clear that the negative patch from one subunit interacts with its own N-terminal domain? This is the impression that one is left with from Fig 2. If it is possible that the C-terminal region interacts with any N-terminal domain within Rad52, this should be pointed out explicitly.

We explicitly show this in the final model. But the point is well taken that Figure 2 does not immediately showcase this difference. We included a statement in the Figure legend so that readers can immediately consider the inter-subunit interactions as well. The statement in the legend reads "While the interactions are portrayed as occurring within a single Rad52 subunit, please note that these interactions also likely occur between two Rad52 subunits in the ring."

(7) Page 12, line 298-299. The word "it" is missing from the sentence that reads "Although the C-tail appears..."

This entire section has been rewritten and edited for clarity.

Reviewer #3 (Remarks to the Author):

Rad52 is an essential protein that promotes homologous recombination by facilitating the formation of Rad51 filaments formation in *S. cerevisiae* by catalyzing the replacement of RPA by Rad51 on ssDNA. To date, the detailed mechanism by which Rad52 promotes Rad51 filament formation has yet to be elucidated. Furthermore, as the C-terminal region of Rad52 is a disordered region, how this region behaves within the Rad52 molecule and affects binding to ssDNA remains enigmatic. This manuscript reports the decameric structure of *S. cerevisiae* Rad52 and provides interesting insights

into the regulation of ssDNA binding by Rad52 via its C-terminal region. However, as commented below, the overall results do not support the assertion of this paper that the decameric Rad52 could serve as a mechanism for the “single-position nucleation” of the Rad51 filament. The authors should reconsider their cryo-EM analyses to improve the quality of the structures obtained and to strengthen their conclusion. At least, the discussion needs to be reconsidered and pursued more carefully. The authors are therefore strongly encouraged to respond to the following comments to improve the overall quality of the manuscript.

We thank the reviewer for the support of our work. We hope the responses address any reservation.

Major Comments:

Comment 1: Regarding the cryo-EM analysis: due to the symmetry of the decameric structure of the N-terminal ring, it may be that the density corresponding to Rad51 converges to a specific position during the 2D classification process, even though Rad51 binds to multiple positions in the Rad52 ring structure. Consequently, the results of their cryo-EM structural analysis are not sufficient to allow the authors to conclude that Rad52 catalyzes Rad51 binding at a single position.

We again thank the reviewer for prompting us to reconsider the model. The single-position binding is not an artifact of the analysis as we would have seen evidence for multi-position binding in the 2D classes. But, there is not a single 2D class that showed even weak density for a second Rad51 binding to the N-terminus.

However, we did not discuss the possibility that the other bound Rad51 molecules exist in another conformation and at a different binding site. The AUC data clearly demonstrates that other Rad51 molecules are bound. The obvious explanation is that there are two modes of Rad51 binding to Rad52: one to the C-terminus and the other to the N-terminal ring. We immediately realized the significance of this finding as it resembles the bipartite Rad51 interactions with the BRC repeats of BRCA2. This would also make sense from an evolutionary perspective as BRCA2 has taken over the Rad51 nucleation functions of Rad52 in higher eukaryotes. The paper has been heavily revised to include this key finding.

Comment 2: In the structural analysis of the full-length Rad52-Rad51 complex, the density observed around the decameric Rad52 could be due to the C-terminal region of Rad52. The authors should improve the analysis to a resolution that would enable a structural model of Rad51 to be fitted.

The overall density we observe for the C-tail is different compared to the Rad51 density in our analysis. The additional data we collected helped better refine the Rad52-Rad51 complex and we have now fit the Rad51 structure into this density. The data are shown in the updated Fig.7B

Comment 3: Supplementary Figure 2: The authors are encouraged to improve the quality of the cryo-EM analysis of Rad52. It seems to me that in the graph, especially the upper

and middle one, the loop of the C-term of the human full length-Rad52 can be vaguely detected. By increasing the number of particles in the analysis, the authors may be able to improve the 2D averaging step to correctly see the C-term.

We have collected and analyzed four different cryoEM datasets for the full-length Rad52 complex. This includes data on different kinds of grids, with and without detergents, data collected with tilt and flipped grids, and finally processed using both Relion and cryoSPARC. The C-tail is too dynamic and exists in non-uniform states for us to reliably deduce any structural information. The presence of the extra density alongside one of the subunits is consistent in all our analysis. We also considered using crosslinkers to improve the structure but were worried about artifacts being introduced given the asymmetry we observe in the untreated samples. Thus, at this point, we are limited to the structure we present in the manuscript.

Minor comments:

Comment 1: P7, lines 176 and 184: "Fig. 2A and B" is the appropriate citation for the figures in the text.

Corrected.

Comment 2: P10, line 249: "Fig. 4A" is the appropriate citation for the figure in the text.

Corrected.

Comment 3: P10, line 257: " Supplementary Fig. 13B" is the appropriate citation for the figure in the text.

Corrected.

Comment 4: Figure 3B appears to be a modified figure of the graph in the first row and second column of Supplementary Figure 6. However, the error bars have been removed and must be added.

Corrected.

Comment 5: In the schema shown in Figure 6, The DNA to which Cy5 and Cy3 have been added does not correspond to the fluorescent molecules attached to the oligos in Supplementary Table 3, which needs to be corrected.

Corrected.

Comment 6: In Figure 6B, the illustration of the C-terminal region of RAD52 needs to be corrected because it overlaps the graph below.

Corrected.

Comment 7: Figure 6E: the authors must specify what the FRET states (1, 2, 3 and 4) correspond to.

We added lines defining the four states in the Figure.

Comment 8: On P.25 in the Materials and Methods section, some characters have been replaced by white squares. Please correct them.

This does not appear to be an issue on our end. We will double check with the journal to solve the issue with PDF conversion.

REVIEWERS' COMMENTS

Reviewer #1 (Remarks to the Author):

I thank the authors for their efforts in addressing the comments. I think the manuscript is now much better: the authors added additional data, and discuss limitations where clearer conclusions have not been possible. The text also reads better. I also agree that the topic remains to be very complex and mutant analysis was too much to ask. I am happy to support the manuscript for acceptance.

Reviewer #2 (Remarks to the Author):

The revisions have satisfied my prior concerns and I am pleased to accept the manuscript.

Reviewer #3 (Remarks to the Author):

In the discussion section of the revised manuscript, the authors have included a new hypothesis, namely that Rad52 may have two Rad51 binding modes (BRCA2-like bipartite Rad51 binding), to their original hypothesis, which is that Rad52 would promote Rad51 filament formation in a single position. The original conclusions have been well revised in this new version of the manuscript, and an overall more interesting and satisfying discussion that focuses on the functional similarities between human BRCA2 and yeast Rad52 in Rad51 loading onto ssDNA is presented. However, despite the authors' efforts to analyze additional cryo-EM data sets, unfortunately this did not help to improve the cryo-EM map of the Rad52-Rad51 complex. Given that the density around the Rad52 ring remains at low resolution in the revised manuscript, and that the binding mode of Rad51 to Rad52 is still unclear, fitting a model structure as in Figure 7B is not appropriate. There is still not enough data to demonstrate that the extra density observed around the Rad52 ring is indeed Rad51.

Response to reviewer comments.
Our responses are in blue.

We thank the reviewers for accepting our revisions.

REVIEWERS' COMMENTS

Reviewer #1 (Remarks to the Author):

I thanks the authors for their efforts in addressing the comments. I think the manuscript is now much better: the authors added additional data, and discuss limitations where clearer conclusions have not been possible. The text also reads better. I also agree that the topic remains to be very complex and mutant analysis was too much to ask. I am happy to support the manuscript for

acceptance.

Thank You for your suggestions and time.

Reviewer #2 (Remarks to the Author):

The revisions have satisfied my prior concerns and I am pleased to accept the manuscript.

Thank You for your suggestions and time.

Reviewer #3 (Remarks to the Author):

In the discussion section of the revised manuscript, the authors have included a new hypothesis, namely that Rad52 may have two Rad51 binding modes (BRCA2-like bipartite Rad51 binding), to their original hypothesis, which is that Rad52 would promote Rad51 filament formation in a single position. The original conclusions have been well revised in this new version of the manuscript, and an overall more interesting and satisfying discussion that focuses on the functional similarities between human BRCA2 and yeast Rad52 in Rad51 loading onto ssDNA is presented.

Thank You for your suggestions and time.

However, despite the authors' efforts to analyze additional cryo-EM data sets, unfortunately this did not help to improve the cryo-EM map of the Rad52-Rad51 complex. Given that the density around the Rad52 ring remains at low resolution in the revised manuscript, and that the binding mode of Rad51 to Rad52 is still unclear, fitting a model structure as in Figure 7B is not appropriate. There is still not enough data to demonstrate that the extra density observed around the Rad52 ring is indeed Rad51. Higher resolution structures will be required to better define the details of the interaction between the two proteins.

We would dearly loved to have higher resolution structures of the complex, but given the dynamic nature of the complex we are left to dissect what the system provides us. The density seen in the Rad52 structure is very different from the Rad52-Rad51 dataset. The helical nature of the Rad51 dimer is unmistakable and the density well agrees with our docking. The data also agrees with the AUC analysis and deletion of the Rad52 C-tail leads to loss in this Rad51 density. The next study will focus on mutagenesis and in-depth investigation of the contacts observed. We present the data with the necessary caveats, and we have included clear statements (please see above) in the Legend for both Figure 7 and Figure 8.